# ICE-Coder: Integrating White-box and Black-box Testing in Execution-guided Multi-agent Code Generation

## Abstract

LLM-based coding agents are programs that utilise LLMs to automate code generation tasks. Typically, they incorporate code execution capabilities which, together with automated test generation and/or debugging methods, enhance the reliability of the generated code. However, the effectiveness of these approaches remains limited in complex problems (such as competitive programming problems) where bugs surface only in convoluted edge cases. This work builds upon multi-agent code generation techniques which emulate software engineering environments. In particular, to address obscure edge cases, we take inspiration from code coverage tools and code reviews to generate white-box tests, on top of existing black-box test generation approaches. Test case outputs are validated through a process of deliberation using the LLM. By increasing the quantity and quality of the test cases, we obtain more reliable generated code. We evaluated ICE-Coder on LiveCodeBench-Hard. Out of the 90 problems, it solves 72, compared to the baseline of 55.

## 1 Introduction

Code generation has emerged as a critical application of large language models (LLMs) (Zan et al., 2023; Jiang et al., 2024; Huynh & Lin, 2025). Recent work underscores this: general-purpose LLMs are often assessed on code generation benchmarks such as HumanEval (Chen et al., 2021a) and LiveCodeBench (Jain et al., 2025), while prompt engineering techniques such as Chain-of-Thought (Wei et al., 2022), Reflexion (Shinn et al., 2023) and Language Agent Tree Search (Zhou et al., 2024) demonstrate their effectiveness in enhancing LLM-based code generation.

Building on these ideas, *LLM-based coding agents* (Hong et al., 2024; Huang et al., 2024b; Islam et al., 2024; Zhong et al., 2024; Dong et al., 2024; Qian et al., 2024; Wang et al., 2024; Shi et al., 2024) have been developed to specifically target the task of code generation. These typically improve upon existing prompt engineering strategies by combining them with testing and debugging techniques.

The success of these approaches demonstrates the crucial role that *testing* plays in automated code generation.

More broadly, two key paradigms have arisen in LLM-based coding agents — (1) to better mirror the way humans behave when they code (Zhong et al., 2024; Huang et al., 2024a; Wang et al., 2024; Shi et al., 2024), and (2) to emulate a software engineering team using multiple agents (Huang et al., 2024b; Islam et al., 2024; Dong et al., 2024; Qian et al., 2024).

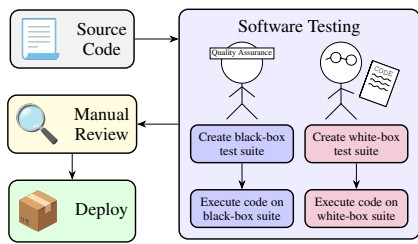

Figure 1: A typical software testing pipeline. The code is tested on black-box and white-box (coverage-based) test cases, after which it must pass a manual check by a reviewer before being deployed.

While many multi-agent frameworks simulate test generation agents, they do not fully simulate the extensive testing done in mature software engineering

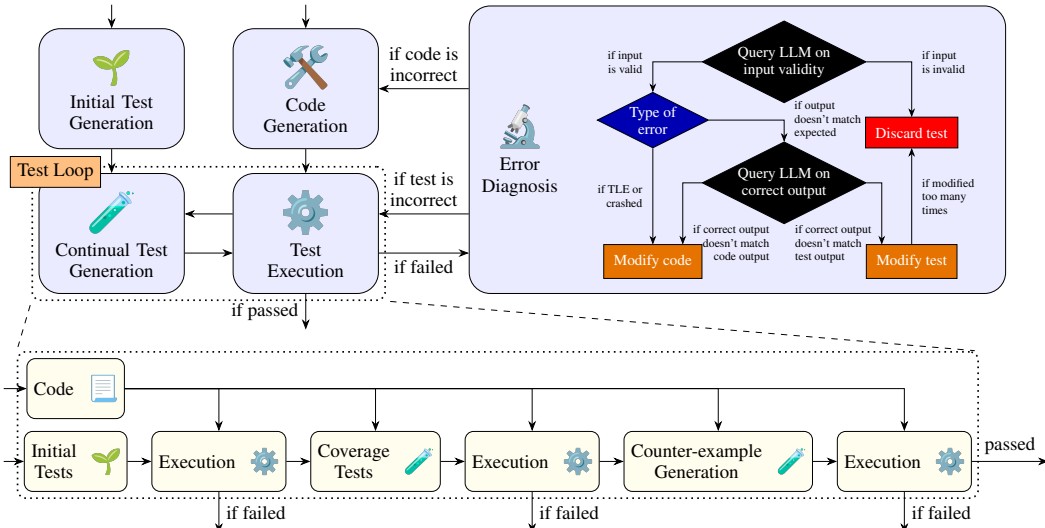

Figure 2: ICE-Coder's pipeline. ICE-Coder cycles between code generation, testing, and error diagnosis. The testing loop is inspired by the software testing pipeline shown in Figure 1.The initial tests correspond to the black-box test cases written by test engineers; the coverage tests emulate the white-box test cases; the counter-example generation mirrors the code review. The code is executed on each of these tests, and tests that fail are passed to the error diagnosis component. Here, by querying the LLM, we decide if the code should be modified, the test case should be modified, or the test case should be discarded. See Figure 4 in Appendix C for a simplified end-to-end execution example.

teams. For example, on top of the black-box tests written by test engineers, developers are often required to write their own *white-box tests*, which aim to maximise *code coverage*. Large companies like Google have sophisticated infrastructures (Ivankovic et al., 2019) to track, manage, and enforce code coverage. Additionally, code must typically undergo *code review* before being accepted (Sadowski et al., 2018), where other developers examine the code to look for bugs and other issues.

These white-box testing steps are crucial for effective bug detection. For particularly challenging problems, the absence of insight into the approach prevents black-box tests from identifying the often subtle or unexpected edge cases it involves, making these white-box tests all the more important. Therefore, we propose simulating coverage-based testing and code reviews in our coding agent.

Another challenge that existing coding agents face is in validating the correctness of generated test case outputs. The expected outputs generated together with test cases are frequently incorrect (Chen et al., 2023; Huang et al., 2024b). Existing coding agents that generate test cases either rely on the (potentially incorrect) expected outputs generated (Huang et al., 2024b), or do consensus voting on the output among the generated codes (Chen et al., 2023). The former approach introduces false positives and the latter approach potentially introduces false negatives for commonly overlooked bugs. For challenging problems in particular, many candidate solutions may fail to solve the same (important) edge case. To address these, we instead propose a *prompt-based approach* to validating test case correctness, similar to how software engineers will deliberate on the correct outputs when the code output differs from the test case output.

In this work, we introduce ICE-Coder, which:

1. combines the two approaches of emulating the software engineering environment and simulating how individual team members think,

2. adds coverage-based testing and "code review" simulations on top of existing black-box testing, and

3. resolves conflicting test case outputs and code outputs through deliberation similar to a software engineering team.

Our approach provides better confidence in the generated code by increasing both the *quantity* and *quality* of test cases.

We evaluated ICE-Coder on LiveCodeBench-Hard on nine LLM models, which differ in size and reasoning ability. ICE-Coder improved the pass@1 rate for all models except GPT 4.1-nano, with the greatest improvement in DeepSeek V3 (59.26% from 27/90 to 43/90) and Grok 3-mini (high) (52.8% increase from 36/90 to 55/90). The best overall solve rate of 72/90 was obtained with GPT o4-mini (high), for which the base model solved just 55/90 problems (thus, we obtained a 30.9% increase). We further investigated the poor performance of GPT 4.1-nano, and posit that it is primarily due to the large initial test generation. Additionally, we compared ICE-Coder with other agentic code generation frameworks, and show that it achieves the state-of-the-art results on LiveCodeBench-Hard.

## 2 RELATED WORK

**LLM-based Coding Agents**  Code generation has emerged both as an important application of LLMs, and as a useful benchmark for the development of LLMs. Apart from the direct training techniques used on the LLMs themselves, several prompt engineering and/or multi-agent frameworks have been developed to enhance the effectiveness of pre-trained LLMs at code generation tasks by incorporating code generation. At their core, these methods rely on iteratively refining the code based on feedback, as is the central thesis of Madaan et al. (2023). Later methods additionally integrate testing or debugging capabilities. Broadly speaking, these can be divided into two categories — those that use multi-agent collaboration to simulate roles in a software engineering team (Hong et al., 2024; Huang et al., 2024b; Islam et al., 2024; Dong et al., 2024; Qian et al., 2024), and those that use prompt engineering techniques on a single LLM agent to have it reason about the problem like a developer (Zhong et al., 2024; Huang et al., 2024a; Wang et al., 2024; Shi et al., 2024).

In Huang et al. (2024b), a separate code generation and test generation agent are used. They also include a test executor agent, though it does not make queries to an LLM. The test executor agent is tasked with generating basic, edge, and large test cases. Generated codes are run on the generated cases, and iteratively refined until they pass all test cases. Dong et al. (2024) uses an analyst, coder, and tester, where the analyst breaks the task down into simpler subtasks for the coder. Similarly, Islam et al. (2024) uses 4 agents, Hong et al. (2024) uses 5 agents and Qian et al. (2024) uses 7 agents, each incorporating finer-grained planning steps. A crucial component of all of these methods is test generation — each of them has a dedicated test generation agent that queries an LLM for test cases.

However, as was studied by Chen et al. (2023); Huang et al. (2024b), the test cases generated by LLMs are unreliable. Chen et al. (2023) proposes a kind of consensus approach, where test cases are deemed accurate when several generated codes yield its expected output. Conversely, Huang et al. (2024b) simply relies on the independence between test and code generation to produce accurate tests, settling for an 89.6% test accuracy rate on HumanEval when using GPT-4. However, such approaches are ineffective for challenging tasks such as competitive programming problems. In these problems, (1) deriving the answer based on the inputs is generally difficult, potentially resulting in the initially generated expected output being incorrect, and (2) there may be important edge cases that many candidate solutions get incorrect. Therefore, we propose (1) using a Chain-of-Thought approach to solicit correct outputs from the LLMs whenever there are disagreements between the code output and expected output, and (2) combining the black-box test generation methods used by AgentCoder with white-box and coverage-based testing methods to derive potentially useful edge cases.

In Zhong et al. (2024), the authors identify a fixed number of code blocks to track the execution state of the code before and after. Shi et al. (2024) introduces a hierarchical "sub-function → file → project" structure for tracing. The traces are then used to identify buggy code blocks for iterative refinement. However, these approaches assume that the high-level structure of the existing code is correct, and that only minor edits to certain code blocks are needed to fix the bugs. We found that this assumption may not be true for more challenging problems. Instead, inspired by their methods, we took a less structured approach to generating execution traces and debugging, allowing the LLMs more freedom to identify the type of bug in the code.

**LLM-based Test Generation**    Besides their applications in collaborative coding agents, the use of LLMs to test existing codebases is itself a subject of independent interest. Early work (Yuan et al., 2024; Chen et al., 2024) simply prompted LLMs to emit tests and then measured how well they compiled and executed. Later work combines LLMs with traditional test generation methods that aim to maximise *code coverage* — the set of lines, or combinations of lines, which were executed. Popular measures of code coverage are line coverage, statement coverage, branch coverage, and path coverage, which differ in level of granularity. Early work relied on constraint solvers to resolve branch conditions and generate tests; this approach is known as *symbolic testing* (King, 1976; Howden, 1977). This was later combined with concrete execution to give *concolic execution* (Godefroid et al., 2005; Sen & Agha, 2006; Cadar et al., 2008; Sapra et al., 2013). Coverage-guided test generation is also used as the fitness measure with evolutionary/genetic algorithms to give mutational/generational test generation tools (Zalewski, 2010; Fraser & Arcuri, 2011; Liu & Yu, 2022).

Yet, symbolic constraint-solving remains challenging, especially in dynamically-typed general-purpose languages like Python, due to the large variable space and frequent use of external libraries (Sapra et al., 2013; Chen et al., 2021b).

Recently, several approaches attempt to use LLMs to solve for inputs that maximise code coverage (Pizzorno & Berger, 2025; Gu et al., 2025; Pan et al., 2025; Ryan et al., 2024). For example, Pizzorno & Berger (2025) first breaks the code under test down into code segments, and derives a set of "interesting" lines — one from each segment. In each prompt, it lists the still-uncovered "interesting" lines based on the current test suite, and iteratively queries the LLM for new tests to add to the test suite. Pan et al. (2025) generates an initial test suite by prompting the LLM, fixes compilation errors in the generated tests, then expands the test suite by prompting the LLM to generate tests for statements that are not already covered. A survey of 161 professional software engineers found a preference for Aster's test cases of those generated by other tools, with 88% of responses stating that the tests generated for Python could be added to a regression suite with no or only minor changes.

Given the role that test generation plays in multi-agent code generation tools, we believe that incorporating such coverage-guided test generation techniques in coding agents is a promising approach to boosting their effectiveness.

## 3    METHODS

Figure 2 illustrates the overall pipeline, consisting of code generation, initial test generation, a test loop, and error diagnosis. The problem is first presented to the code generator and initial test generator respectively. Then, the generated code is executed in the test loop, which also generates additional white-box tests on top of the initial black-box ones. If the output of the code does not match the expected output of a given test, we perform "error diagnosis" to determine if the code or the test case has to be changed, or if the test case is malformed. Then, we return to the test loop after making appropriate changes.

### 3.1    CODE GENERATION

**Initial Code Generation**    To maximise the chances of generating correct code, we use Chain-of-Thought (CoT) prompting, specifically directing the LLM to focus on correctness and efficiency. The LLM is asked to (1) understand the task, (2) relate the task to similar problems, (3) analyse various methods to solve the problem, (4) analyse the time complexity, (5) choose the most efficient, correct method, (6) write the solution in pseudocode, and (7) generate annotated Python code. The detailed prompt templates are provided in Appendix H.

**Code Refinement**    When modifying incorrect code, we first record the execution trace (local variable changes for each line of execution) of the failed code on the test case, which provides the LLM with more information on the execution. To keep within the token limit, we record a shortened execution trace by only printing the environment the first 1-2 times each line is executed. Since many of these lines are executed consecutively, we simply log the change in variable values between consecutive lines, rather than the entire context. Then, we pass this execution trace, along with the failed code and test case, to a separate LLM to generate a reflection on why the code is wrong. We find that this reflection step prevents non-reasoning LLMs from creating a single if condition to deal

with the specific test case, instead encouraging the LLM to generalise the problem. Thereafter, we pass the old code and the reflection to the LLM to generate a new piece of code, using a similar CoT prompt. In the case that the LLM never finds a correct solution, we limited the number of code regenerations to eight to prevent ICE-Coder from running indefinitely.

## 3.2 INITIAL TEST GENERATION

The **initial test set** is generated independently from the code. This makes it impervious to spillover biases from the code. They are an effective first line of defence, and are especially useful in flagging out codes that misinterpret the problem, or that completely overlook a scenario. It is also at this stage that we generate large-scale test cases to stress-test the code. The initial test set is obtained in a similar way as AgentCoder (Huang et al., 2024b) — by prompting the LLM to generate basic, edge and large-scale test cases. Unlike AgentCoder, we dedicate one prompt for each type of test case, thus yielding a larger initial test set. For edge cases, we first prompt the LLM to generate a list of edge scenarios for the problem, then prompt the LLM for test cases for each edge scenario. Especially for large and/or reasoning models, we found that increasing the number of test cases in this way is effective in ensuring that each functional scenario is tested and catching obscure edge cases. See Algorithm 2 in Appendix B for the details.

The LLMs were allowed to generate input/output pairs either as strings or as Python expressions. If Python expressions are generated, they are evaluated and piped as inputs to the generated code during test execution. Allowing the generation of Python expressions (1) helps to save tokens when generating large-scale test cases, and (2) avoids the frequent counting errors made by LLMs when generating test inputs.

## 3.3 TEST LOOP

---

**Algorithm 1** Test Loop

---

**Inputs:**     Problem statement $P$, Code $\sigma$, Test Suite $\psi$
**Outputs:**     Counter-example $\tau'$, Updated Test Suite $\psi'$

1: statementsLeft $\leftarrow$ GETEXECUTABLESTATEMENTS$(\sigma)$
   $\overline{\text{Existing tests}}$ *(initial test set + any other tests generated previously)*
2: **for** $\tau \in \psi$ **do**
3:     testResult, statementsRun $\leftarrow$ execute$(\sigma(\tau.\text{inputs}))$
4:     **if** statementsRun $\setminus$ statementsLeft $\neq \emptyset$ **then** $\psi$.moveToFront$(\tau)$ **end if**
5:     statementsLeft $\leftarrow$ statementsLeft $\setminus$ statementsRun
6:     **if** testResult is ERROR $\vee$ testResult $\neq \tau$.expected **then**
7:         $\psi$.moveToFront$(\tau)$
8:         **return** $\tau, \psi$
9:     **end if**
10: **end for**
    $\overline{\text{Coverage-guided testing}}$
11: **while** statementsLeft $\neq \emptyset$ **do**
12:     $\rho \leftarrow$ statementsLeft.getOneElement()
13:     statementsLeft $\leftarrow$ statementsLeft $\setminus \{\rho\}$
14:     $\tau \leftarrow$ PROMPTLLM(format(generateTestForLine, $P, \sigma, \rho.\text{lineNum}$))
15:     $\psi$.addToFront$(\tau)$
16:     testResult, statementsRun $\leftarrow$ execute$(\sigma(\tau.\text{inputs}))$
17:     statementsLeft $\leftarrow$ statementsLeft $\setminus$ statementsRun
18:     **if** testResult is ERROR $\vee$ testResult $\neq \tau$.expectedOutput **then return** $\tau, \psi$ **end if**
19: **end while**
    $\overline{\text{Counter-example generation}}$
20: $\tau \leftarrow$ PROMPTLLM(format(generateCounterexample, $P, \sigma$))
21: $\psi$.addToFront$(\tau)$
22: testResult, statementsRun $\leftarrow$ execute$(\sigma(\tau.\text{inputs}))$
23: **if** testResult is ERROR $\vee$ testResult $\neq \tau$.expected **then return** $\tau, \psi$ **end if**
24: **return** NONE, $\psi$

---

Three types of tests are run in the test loop — (1) the initial black-box tests, (2) coverage-guided white-box tests, and (3) direct queries for counter-examples to the code. The test suite is executed in order. Whenever a test case covers a new line of code or discovers a bug, it is moved to the front of this linked list. Thus, test cases most likely to discover bugs are prioritised. Newly generated tests are added to the front of the linked list. See Algorithm 1 for the full details on test execution.

The test executor relays a counter-example input (if found) to the error diagnosis component. If the test case is found to be incorrect or invalid, the test executor can continue from where it left off. Besides that, even after finding a valid counter-example to the code, the main logic will continue running the test executor if not all statements have been covered. However, any subsequent counter-examples returned by the test executor will not be deliberated on.

**Coverage-guided tests** When executing the code, we aim for complete statement coverage. This ensures that each statement behaves as expected. For example, statements involving floating point operations may unexpectedly generate spurious digits in an otherwise logically-sound code. This might only be discovered by executing that specific statement. Additionally, keeping and re-using coverage-guided test cases helps us avoid regression when refining codes by ensuring that previously accounted-for cases are still handled correctly. Thus, for statements not covered after executing the existing test suite, we prompt the LLM with the problem, the code, and each of their line numbers, and ask the LLM to generate tests that target each of those specific lines. See Appendix E for an example of how a coverage-guided test case was used to expose a bug.

**Counter-example generation** Finally, if it had passed all prior tests, we directly query the LLM to generate a single counter-example to the code as a final check. Similar to a code review, this directly checks if there is anything the code might have missed. See Appendix F for an example of how a coverage-guided test case was used to expose a bug. To prime the LLM to find a counter-example, we tell it that there is a bug in the code and ask it to find a test case that exposes the bug. See Appendix H for the prompt templates used.

## 3.4 Error Diagnosis

We perform **error diagnosis** whenever either (1) the code output does not match the expected output of the test case, (2) the code crashes on the test case, or (3) the code exceeds the time limit on a test case. In case 1, we query a separate LLM on which output is correct.

In all cases, we first verify if the input formatting is correct by querying the LLM. Generated tests sometimes have counting errors with parameters (e.g. the test claims to input a 20-character string, but only produces a 19-character string). Sometimes, generated tests do not fit within the parameter constraints for the size of input variables (e.g. sets an input to 0 when the problem states that it cannot be 0), or do not correspond to a valid instance of the problem, thereby having no solution (e.g. produces a general graph for a problem where the input should be a tree). This first check seeks to address these problems. If the inputs are invalid, the test case is discarded entirely.

Otherwise, if the time limit was exceeded (TLE) or it crashed, we return the code to the code generator for modification. If the code output does not match the expected output, we additionally query the LLM on what the correct output should be, we then modify the test case expected outputs and/or the code based on the result of this query. (Due to the difficulty of reasoning about large test cases, we do not check the code outputs against the expected outputs for large test cases. In other words, large test cases are mainly used to test the efficiency of the code.)

Finally, if the expected output of a test case had been modified more than once, this suggests that the solution to the problem instance is not obvious to the LLM. In such instances, we simply discard the test case. Also, to balance token usage with accuracy, we limit the number of times error diagnosis is conducted to 16. Upon crossing this limit, we simply return the code as is.

## 4 Evaluation

### 4.1 Experimental Setup

We evaluated ICE-Coder against the latest version (August 2024) of LiveCodeBench-Hard (Jain et al., 2025), which is a popular dataset for competitive programming problems that includes prob-

lems from CodeForces, LeetCode, and AtCoder. We chose LiveCodeBench for evaluation as it draws problems from a wide variety of sources. As current models are already able to solve most of the easier problems, we specifically targeted the hard section of this competitive programming dataset. We used the pass@1 metric for evaluation. The generated codes were run on our server (Intel(R) Xeon(R) Gold 6248R CPU @ 3.00GHz) with a time limit of 6 seconds[1] for each test case.

First, ICE-Coder was run on several different popular LLMs to test its LLM-agnosticism. The LLMs — Gemini-2.5 Pro, DeepSeek V3, DeepSeek R1, Grok 3-mini, GPT 4.1-nano, GPT 4.1-mini, and GPT o4-mini — were chosen to (1) reflect a variety of different providers, (2) test out different types of models (e.g. reasoning/non-reasoning, mini/larger), and (3) for good baseline performance on the LiveCodeBench leaderboard. The exact checkpoints of the models used are detailed in Appendix A for reproduction. For the baseline, we used the initial code generation prompt template (provided in Appendix H) to generate the codes. Since we automated the extraction of codes, we allowed the LLMs at most 3 attempts to output codes in that format (the first attempt conforming to the format was taken to be the submission).

Next, we compared ICE-Coder against other LLM-based agentic code generation tools — Agent-Coder (Huang et al., 2024b), MapCoder (Islam et al., 2024), LDB (Zhong et al., 2024), and LPW (Lei et al., 2025). These represent a variety of different methods. AgentCoder was picked as the prompts used in ICE-Coder were adapted from it. Conversely, LDB showcases a debugging-based method to improve code generation. Additionally, LDB and LPW were picked for good performance on HumanEval (Chen et al., 2021a), and MapCoder was picked for good performance on MBPP[2] (Austin et al., 2021). Unfortunately, we could not find other open-source agentic tools that target the LiveCodeBench dataset. For tools that generate test cases, we additionally measure the code coverage of the generated test cases on the generated codes. Unfortunately, since LiveCodeBench does not provide canonical solutions, we cannot measure test case accuracy. The tests were conducted with GPT o4-mini (high) as the base model, since it is the best-performing base model on the LiveCodeBench leaderboard at the time of writing (though notably, when we performed our own benchmarking, Gemini-2.5 Pro and DeepSeek R1 performed better).

## 4.2 RESULTS

As shown in Table 1, ICE-Coder improves the performance of all the models tested on LiveCodeBench-Hard, apart from GPT 4.1-nano (see Appendix D for analysis). The best model on which it performs is GPT o4-mini (high), and the steepest improvement is seen with DeepSeek V3.

As shown in Table 2, ICE-Coder achieves state-of-the-art results (80.00% or 72 out of 90) when used on GPT o4-mini (high). Additionally, it achieves the highest average code coverage of 97.82%.

Table 1: pass@1 Rates of ICE-Coder on LiveCodeBench-Hard

| Model | Baseline | ICE-Coder | Improvement |
|---|---|---|---|
| Gemini-2.5 Pro | 65.56% | 74.44% | 13.56% |
| DeepSeek V3 | 30.00% | 47.78% | 59.26% |
| DeepSeek R1 | 62.22% | 74.44% | 19.64% |
| Grok 3-mini | 36.67% | 45.56% | 24.24% |
| Grok 3-mini (high) | 40.00% | 61.11% | 52.78% |
| GPT 4.1-nano | 7.78% | 5.56% | -28.57% |
| GPT 4.1-mini | 28.89% | 38.89% | 34.62% |
| GPT o4-mini | 60.00% | 77.78% | 29.63% |
| GPT o4-mini (high) | 61.11% | 80.00% | 30.91% |

---

[1] The time limit of 6s is based on the LiveCodeBench implementation on GitHub.

[2] Since we are only testing open-source agentic methods, other methods on the MBPP leaderboard such as EG-CFG (Lavon et al., 2025) and QualityFlow (Hu et al., 2025) were not included in our comparison.

[3] If no syntactically-valid code is produced, the code coverage for that problem is marked as 0%.

[4] We attribute the lower-than-baseline pass@1 rate of AgentCoder to the prompt used — the prompt used for the baseline mentioned competitive programming, thus priming the LLM to generate more efficient code.

[5] LPW is made for functional problems. The codes produced by LPW contain many print statements for debugging. During evaluation, we replaced these print statements with pass statements. Had we not replaced these print statements and merely suppressed them, this pass@1 rate drops to 28.89% due to time limit exceeded errors. As for the results, we posit that the planning workflow proposed by LPW is less effective on a reasoning model like GPT o4-mini.

Table 2: Results on LiveCodeBench-Hard with GPT o4-mini

| Framework/Tool | Average Code Coverage[3] | pass@1 rate |
|---|---|---|
| Baseline | NA | 61.11% |
| AgentCoder | 92.43% | 60.00%[4] |
| MapCoder | NA | 71.11% |
| LDB | 92.31% | 78.89% |
| LPW | 82.29% | 54.44%[5] |
| ICE-Coder | 97.82% | 80.00% |

## 4.3 ANALYSIS

We divide our analysis based on whether the initial code generated is correct.

If the initial code is already correct, the goal is to avoid regression. Ideally, we want the code to pass all the generated test cases and be returned. While this happens most of the time, incorrectly produced test cases do complicate things for some of the problems. We've attempted to use detailed CoT (see Appendix H) to mitigate this in the diagnosis phase. The blanket input validity check was also inspired by regressions during our experimentation, where minor formatting errors (e.g. presence or absence of spaces after commas) in the inputs did not crash the code, but affected the diagnosis results (e.g. both should not have spaces after commas, but since the input incorrectly contained spaces, the diagnosis favoured the output with the spaces). Nevertheless, there were still incorrect test cases that get through the discussion phase, especially in calculation-heavy questions and non-reasoning models. This is partially mitigated by our provision of the execution trace in the reflection stage, where the LLM might realise based on the trace that the code is actually correct, and the expected output is not. Still, there remain issues with LLMs being overly trusting of the expected outputs claimed by the prompt. Another potential solution is to equip the LLMs with a calculator tool, e.g., using a ReAct framework (Yao et al., 2023). We leave this to future work.

See Appendix G for an example of regression with DeepSeek V3. These "bad" test cases are most frequently contributed by the initial test generation, where each LLM response yields multiple test cases at once. Such scenarios also explain the poor performance of ICE-Coder on GPT 4.1-nano (see Appendix D for more).

If the initial code is incorrect, we need to (1) find a test case that demonstrates why the code is incorrect, and (2) correct the code. As demonstrated by our ablation study in Section 4.4, each component of ICE-Coder is essential in this process. In Appendices E and F, we detail examples where coverage-guided tests and counter-example generation were able to find bugs. In general, the large initial black-box test set serves as an effective first line of defence against bugs. However, especially when the solution code is complex, the initial test set is often insufficient in exhausting all branching scenarios. In these cases, coverage-guided tests and counter-example generation supplement and complement the initial test set. Finally, the execution traces help with diagnosing the problem and correcting the code.

## 4.4 ABLATION STUDY

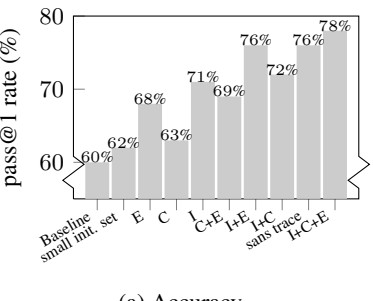

(a) Accuracy

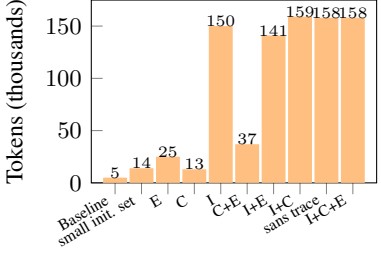

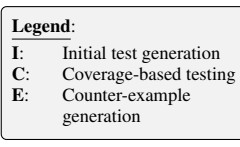

(b) Median token cost

Figure 3: Ablation study with GPT o4-mini

We conducted an ablation study on the three types of tests generated — the initial black-box test generation, the coverage-guided test generation, and the counter-example generation. We tested all 8 combinations of these test generation methods, using GPT o4-mini as the base model. We also report the token usage for each. We also ran two more experiments — (1) with a smaller initial test set using a single prompt to get basic, edge, and large test cases (similar to AgentCoder)[6], and (2) without producing the execution trace for the reflection step[7]. The results of the ablation study on the testing methods are shown in Figure 3.

As shown in Figure 3a, the pass@1 rate increases with each type of test added. On their own, each type of test is able to achieve significant improvements over the baseline, with the most significant improvement coming from the initial test generation. Yet, at the same time, the initial test generation uses by far the greatest number of tokens. (Note: for some of the tests, adding a component on top of the initial test generation decreased the total token usage; this is likely due to randomness and variance in the initial test generation, which dominates the token usage.)

Conversely, coverage-guided testing and counter-example generation add minimal token overhead to the code generation process whilst improving the pass@1 rates. When compared more fairly to a smaller initial test set (with similar token usage), they achieve better pass@1 rates.

Similarly, the addition of tracking of execution traces adds negligible token overhead whilst improving accuracy rates. In fact, while the execution traces increase the number of input tokens used (from 59K to 67K median), they slightly decrease the expenditure of reasoning tokens (from 84K to 76K median). (The median response tokens are around the same for both — about 15K.)

## 5 THREATS TO VALIDITY

We only evaluated ICE-Coder on LiveCodeBench-Hard, and have not tested it on other datasets. Each experiment was only conducted once on the relatively small 90-problem dataset, which means some variance should be expected of the results. Additionally, the evaluation was conducted on our Intel(R) Xeon(R) Gold 6248R CPU server, which may have a different execution speed from other evaluation platforms.

We set certain constant limits, e.g. number of code regenerations (8), number of error diagnosis steps performed (16), number of test case updates (1), etc. when doing our experiments. We have not spent too much effort analysing how these constants affect the balance between code accuracy and token cost.

For the comparison with other agentic tools, since they were not tailored for LiveCodeBench, some of them only write Python functions. To test these, we had to slightly modify the LiveCodeBench problem descriptions to make them functional, rather than I/O-based [8]. Thus, while we tried our best to keep the modifications minimal, the results could nevertheless be affected by how we modified the problem descriptions.

## 6 CONCLUSION

In this paper, we introduce ICE-Coder, which improves on previous multi-agent code generation frameworks for complex problems by introducing white-box and coverage-guided tests, and by taking a prompt-based approach to figuring out correct generated test outputs. In doing so, it simulates a typical software engineering pipeline, with black-box functional tests, white-box coverage tests, and code reviews. At the same time, it also mirrors how individual developers work, deliberating on the correct test outputs and using the execution trace to find bugs. ICE-Coder achieves state-of-the-art performance on the LiveCodeBench-Hard dataset, with an 80% pass@1 rate when used with GPT o4-mini (high). It also improves the pass@1 rates on a range of other models, including a 59% improvement on DeepSeek V3 and a 53% improvement on Grok-3 mini (high).

---

[6]This experiment was conducted with the small initial test set as the only source of test cases.

[7]This experiment was conducted with all three types of tests.

[8]We transformed them to `solve(inp: str) -> str` functions, with the problem description as a docstring.

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

## A    MODEL CHECKPOINTS

Where applicable, the exact checkpoints of the models used for benchmarking are detailed here for reproduction.

| Model | Checkpoint |
|---|---|
| Gemini-2.5 Pro | (latest stable release) |
| DeepSeek V3 | DeepSeek-V3-0324 |
| DeepSeek R1 | DeepSeek-R1-0528 |
| Grok 3-mini | (latest stable release) |
| GPT 4.1-nano | gpt-4.1-nano-2025-04-14 |
| GPT 4.1-mini | gpt-4.1-mini-2025-04-14 |
| GPT o4-mini | o4-mini-2025-04-16 |

## B    INITIAL TEST GENERATION ALGORITHM

---
**Algorithm 2** Initial Test Generation

---
**Inputs:**     Problem statement $P$
**Outputs:**    Test suite $\psi$

1: scenarios $\leftarrow$ PROMPTLLM(format(generateEdgeScenarioPrompt, $P$))
2: scenarios.add(BasicScenario)
3: scenarios.add(LargeScenario)
4: TestMap $\leftarrow$ new MAP()
5: **for** scenario $\in$ scenarios **do**
6:     TestMap[scenario] $\leftarrow$ PROMPTLLM(format(generateTestPrompt, $P$, scenario))
7: **end for**
8: TestMap.REMOVEDUPLICATETESTS()
9: $\psi \leftarrow$ INTERLEAVE(TestMap)
10: **return** $\psi$

---

## C EXAMPLE END-TO-END EXECUTION FLOW

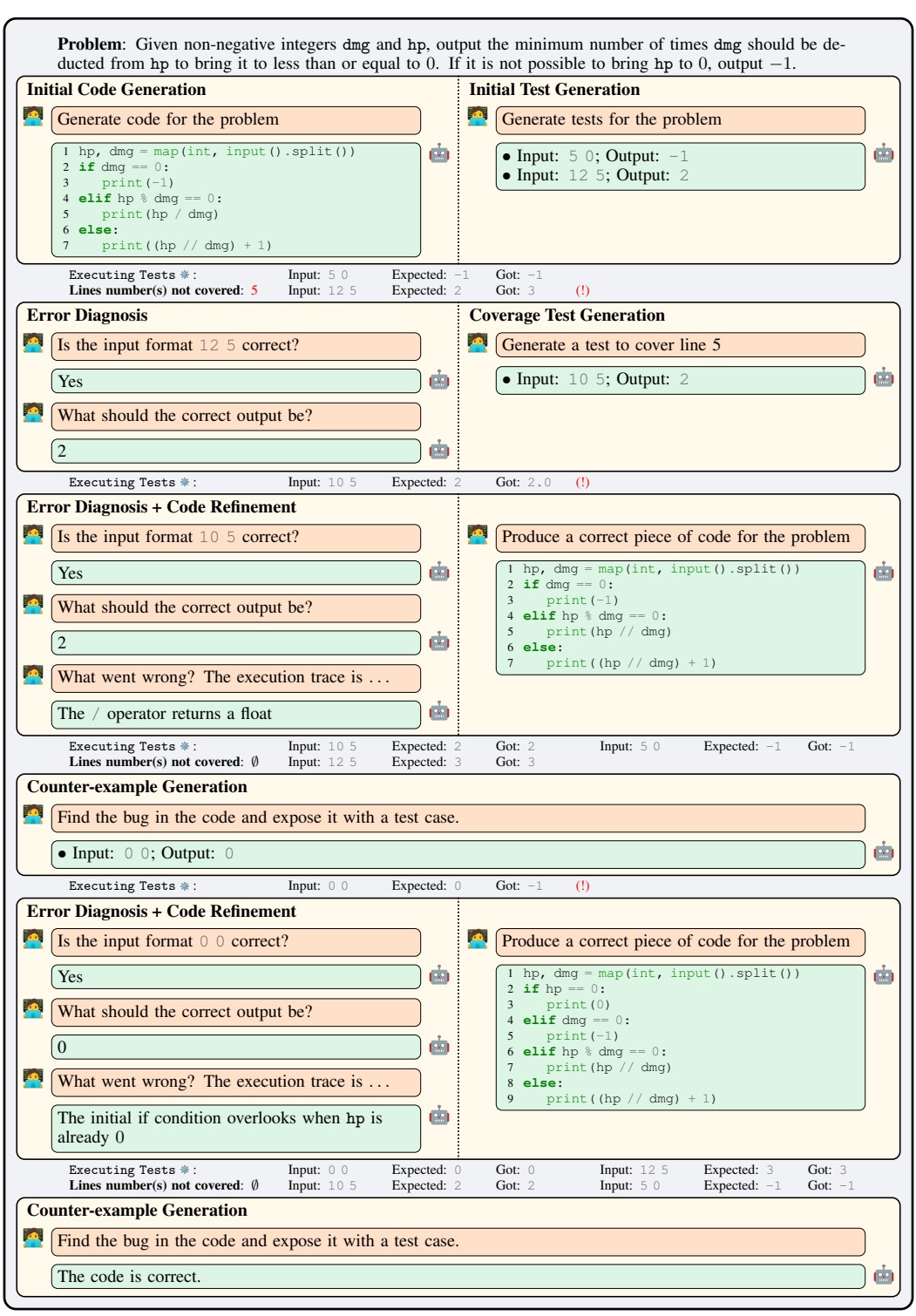

Figure 4: Illustration of the execution flow on an example toy problem. For the actual prompts used, refer to Appendix H.

## D  INVESTIGATING GPT 4.1-NANO ANOMALY

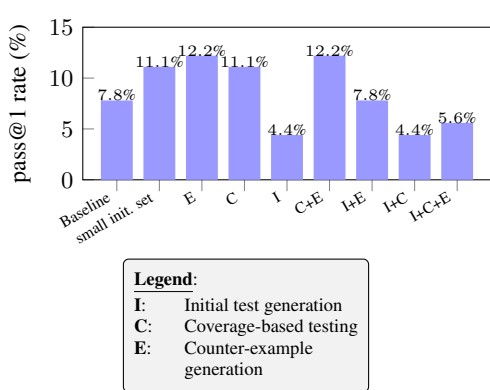

We investigated the anomalous results for GPT 4.1-nano further by running it with each combination of testing components. The results are detailed in Figure 5.

With GPT 4.1-nano, we found that ICE-Coder performs better without the initial black-box test generation. With GPT 4.1-nano, ICE-Coder obtains a 12.2% pass@1 rate without the initial test generation, and only 5.56% when the initial test generation was included. GPT 4.1-nano frequently produced test cases with incorrect expected outputs or formats, which it frequently is unable to identify in the diagnosis/deliberation stage, thus negatively affecting its results. The risk of generating such "bad" test cases is incurred for each test case we generate. Therefore, with GPT 4.1-nano, using a small number of targeted test cases is more effective than using a large initial black-box test

Figure 5: Component Analysis with GPT 4.1-nano

set.

## E  COVERAGE-GUIDED TESTING CASE STUDY

Here, we examine how ICE-Coder found a bug in its candidate solution for the "Cans and Openers" problem (question ID abc312_f of LiveCodeBench taken from AtCoder) due to coverage-guided test generation when using GPT o4-mini.

The "Cans and Openers" problem is reproduced here:

> There are $N$ items.
> Each of these is one of a pull-tab can, a regular can, or a can opener.
> The $i$-th item is described by an integer pair $(T_i, X_i)$ as follows:
>
> - If $T_i = 0$, the $i$-th item is a pull-tab can; if you obtain it, you get a happiness of $X_i$.
> - If $T_i = 1$, the $i$-th item is a regular can; if you obtain it and use a can opener against it, you get a happiness of $X_i$.
> - If $T_i = 2$, the $i$-th item is a can opener; it can be used against at most $X_i$ cans.
>
> Find the maximum total happiness that you get by obtaining $M$ items out of $N$.
>
> Input
>
> The input is given from Standard Input in the following format:
> $N\ M$
> $T_1\ X_1$
> $T_2\ X_2$
> $\vdots$
> $T_N\ X_N$
>
> Output
>
> Print the answer as an integer.

Constraints

- $1 \le M \le N \le 2 \times 10^5$
- $T_i$ is 0, 1, or 2.
- $1 \le X_i \le 10^9$
- All input values are integers.

Sample Input 1

```
8 4
0 6
0 6
1 3
1 5
1 15
2 1
2 10
2 100
```

Sample Output 1

```
27
```

If you obtain the 1-st, 2-nd, 5-th, and 7-th items, and use the 7-th item (a can opener) against the 5-th item, you will get a happiness of $6 + 6 + 15 = 27$.
There are no ways to obtain items to get a happiness of 28 or greater, but you can still get a happiness of 27 by obtaining the 6-th or 8-th items instead of the 7-th in the combination above.

Sample Input 2

```
5 5
1 5
1 5
1 5
1 5
1 5
```

Sample Output 2

```
0
```

Sample Input 3

```
12 6
2 2
0 1
0 9
1 3
1 5
1 3
0 4
2 1
1 8
2 1
0 1
```

0 4

Sample Output 3

30

There is an efficient $O(n \log n)$ solution to the problem. To understand the solution, consider the values $u_p$ and $u_r$, which represent the total satisfaction obtained from the chosen pull-tab cans and from the chosen regular cans respectively. We want to maximise both $u_p$ and $u_r$, but there is a trade-off between them due to the limited number of items we can pick. A Pareto efficient choice of items is one in which we cannot increase either $u_p$ or $u_r$ without decreasing the other, and the Pareto front is the set of all Pareto efficient choices of items. On one end of the Pareto front, we pick the $M$ highest-satisfaction pull-tab cans (and fill in any remaining slots with the best combination of can openers and regular cans). On the other end of the Pareto front, we pick the best combination of can openers and regular cans (and then fill in any remaining slots with the highest-satisfaction pull-tab cans). Note that "the best combination of can openers and regular cans" can be found via a greedy algorithm, where we pick the best can opener if we have no "opens" left, and the best regular can if we do have "opens" left.

The algorithm then traverses the Pareto front, starting on one end, and moving to the other by replacing cans one at a time. In each iteration, a greedy (hence efficient) substitution is made, and the satisfaction updated and recorded. Then, the maximum satisfaction is output. This can be achieved in $O(n \log n)$ via either prior sorting, or using sorted data structures such as binary heaps or binary trees.

Initially, GPT o4-mini outputs a buggy solution which attempts a similar approach. It starts with a set of only pull-tab cans. Then, it iterates through all possible numbers of can openers. In each iteration, it replaces a pull-tab can with a can opener, then greedily swaps pull-tab cans for better regular cans, limited to the available "opens":

```python
import sys
import threading
def main():
    import sys
    import heapq

    data = sys.stdin.read().split()
    N, M = map(int, data[:2])
    idx = 2

    tabs = []
    cans = []
    openers = []

    # Read and bucket items by type
    for _ in range(N):
        t = int(data[idx]);  x = int(data[idx+1])
        idx += 2
        if t == 0:
            tabs.append(x)
        elif t == 1:
            cans.append(x)
        else:  # t == 2
            openers.append(x)

    # We will try k = 0..min(#openers, M)
    openers.sort(reverse=True)

    # Max-heaps for unused tabs and cans (store negatives)
    outs_tab = [-x for x in tabs]
```

```
918        heapq.heapify(outs_tab)
919        outs_can = [-x for x in cans]
920        heapq.heapify(outs_can)
921
922        # picks = min-heap of (x, is_can) for the items currently
923        ↪  counted
924        picks = []
925        cur_sum = 0
926        can_in_picks = 0
927
928        cap = 0        # total capacity from chosen openers
929        t = M          # how many non-opener slots we must fill
930
931        # Initial fill with tabs only (capacity=0 => no cans allowed)
932        while len(picks) < t and outs_tab:
933            x = -heapq.heappop(outs_tab)
934            heapq.heappush(picks, (x, False))
935            cur_sum += x
936
937        ans = 0
938        max_k = min(len(openers), M)
939
940        for k in range(max_k + 1):
941            # Record the best sum for choosing exactly k openers
942            if cur_sum > ans:
943                ans = cur_sum
944            if k == max_k:
945                break
946
947            # 1) Add the k-th opener
948            cap += openers[k]
949            # 2) One more slot is taken by that opener
950            t -= 1
951
952            # 3) Remove the smallest item from picks (since t
953            ↪  decreased by 1)
954            if picks:
955                x_min, is_can_min = heapq.heappop(picks)
956                cur_sum -= x_min
957                if is_can_min:
958                    can_in_picks -= 1
959
960            # 4) Refill picks up to size t
961            while len(picks) < t:
962                # best can if we still have capacity for more cans
963                x_can = None
964                if can_in_picks < cap and outs_can:
965                    x_can = -outs_can[0]
966                # best tab
967                x_tab = -outs_tab[0] if outs_tab else None
968
969                # If no candidate at all, break
970                if x_can is None and x_tab is None:
971                    break

                # pick whichever yields higher X
                if x_can is not None and (x_tab is None or x_can >
                ↪  x_tab):
                    # take can
```

```
                        heapq.heappop(outs_can)
                        heapq.heappush(picks, (x_can, True))
                        cur_sum += x_can
                        can_in_picks += 1
                    else:
                        # take tab
                        heapq.heappop(outs_tab)
                        heapq.heappush(picks, (x_tab, False))
                        cur_sum += x_tab

            # 5) Swap loop to improve sum by swapping in better cans
            ↪  if capacity allows
            while can_in_picks < cap and outs_can and picks:
                x_can = -outs_can[0]      # best unused can
                x_min, is_can_min = picks[0]  # smallest current pick
                if x_can <= x_min:
                    break

                # do the swap
                heapq.heappop(outs_can)
                heapq.heappop(picks)
                cur_sum += (x_can - x_min)
                # the new pick is a can
                heapq.heappush(picks, (x_can, True))

                # put back the old smallest into its proper outs-heap
                if is_can_min:
                    # swapped can → can
                    heapq.heappush(outs_can, -x_min)
                    # can_in_picks does not change
                else:
                    # swapped tab → can
                    heapq.heappush(outs_tab, -x_min)
                    can_in_picks += 1

        sys.stdout.write(str(ans) + "\n")

if __name__ == "__main__":
    main()
```

However, this strategy fails because the pull-tab can is strictly replaced by the can-opener at the start of the iteration, even when there is spare space. Though a while loop is used to add regular cans to the set of picks to fill up spare capacity, this does not work when there are few regular cans. This is caught by a coverage-guided test case for the **if** x_can **is None and** x_tab **is None** condition to break the while loop for refilling picks "If no candidate at all":

```
6 6
0 2
0 3
0 4
1 10
1 9
2 2
```

Here, it is possible to pick all items. Thus, the expected output is 28. However, the code outputs 26, picking only 5 of the items and leaving out the first pull-tab can.

After several iterations of reflection and refinement, the LLM finds the correct solution:

```python
import sys
import threading

def main():
    import sys
    data = sys.stdin.read().split()
    it = iter(data)
    N = int(next(it))
    M = int(next(it))

    tabs = []
    cans = []
    openers = []

    # 1) Read and classify items
    for _ in range(N):
        t = int(next(it)); x = int(next(it))
        if t == 0:
            tabs.append(x)
        elif t == 1:
            cans.append(x)
        else:
            openers.append(x)

    # 2) Sort descending
    tabs.sort(reverse=True)
    cans.sort(reverse=True)
    openers.sort(reverse=True)

    # 3) Prefix sums of opener capacities
    Kmax = min(len(openers), M)
    cap = [0] * (Kmax + 1)
    for i in range(1, Kmax + 1):
        cap[i] = cap[i-1] + openers[i-1]

    import heapq
    sel = []        # min-heap of currently selected items
    other = []      # max-heap (store negatives) of the rest
    sum_sel = 0
    K = M           # current # of slots for tabs+cans

    # DS_insert maintains the top-K items in 'sel' and the rest in
    #    'other'
    def DS_insert(x):
        nonlocal sum_sel
        if len(sel) < K:
            # room to push directly
            heapq.heappush(sel, x)
            sum_sel += x
        else:
            # if x is better than the smallest in sel, swap it in
            if K > 0 and sel and x > sel[0]:
                y = heapq.heappop(sel)
                sum_sel -= y
                # the dropped item goes to 'other'
                heapq.heappush(other, -y)
                # now insert x into sel
                heapq.heappush(sel, x)
                sum_sel += x
```

```
1080                 else:
1081                     # otherwise x is never selected
1082                     heapq.heappush(other, -x)
1083
1084         # 4) Insert all pull-tabs first
1085         for x in tabs:
1086             DS_insert(x)
1087
1088         best = sum_sel
1089         j_inserted = 0
1090
1091         # 5) Sweep over k = number of openers chosen
1092         for k in range(1, Kmax + 1):
1093             # a) unlock and insert newly available cans
1094             j_target = min(cap[k], len(cans))
1095             while j_inserted < j_target:
1096                 DS_insert(cans[j_inserted])
1097                 j_inserted += 1
1098
1099             # b) pay one slot for the k-th opener
1100             K -= 1
1101             # if we have too many in sel, drop the smallest
1102             if len(sel) > K:
1103                 y = heapq.heappop(sel)
1104                 sum_sel -= y
1105                 # the dropped item is gone for good
1106
1107             # c) update the best answer
1108             if sum_sel > best:
1109                 best = sum_sel
1110
1111     # 6) Output result
1112     sys.stdout.write(str(best))
1113
1114 if __name__ == "__main__":
1115     main()
```

## F   COUNTER-EXAMPLE GENERATION CASE STUDY

Here, we examine how ICE-Coder found a bug in its candidate solution for the "maximum-number-of-groups-with-increasing-length" problem (question ID 2919 of LiveCodeBench taken from Leet-Code) during the counter-example generation phase when using GPT o4-mini.

The "maximum-number-of-groups-with-increasing-length" problem is reproduced here:

> You are given a 0-indexed array usageLimits of length $n$.
> Your task is to create groups using numbers from 0 to $n - 1$, ensuring that each number, $i$, is used no more than usageLimits[i] times in total across all groups. You must also satisfy the following conditions:
>
> Each group must consist of distinct numbers, meaning that no duplicate numbers are allowed within a single group.
> Each group (except the first one) must have a length strictly greater than the previous group.
>
> Return an integer denoting the maximum number of groups you can create while satisfying these conditions.

Example 1:

Input: `usageLimits` $= [1, 2, 5]$
Output: 3
Explanation: In this example, we can use $0$ at most once, $1$ at most twice, and $2$ at most five times.
One way of creating the maximum number of groups while satisfying the conditions is:
Group 1 contains the number $[2]$.
Group 2 contains the numbers $[1, 2]$.
Group 3 contains the numbers $[0, 1, 2]$.
It can be shown that the maximum number of groups is 3.
So, the output is 3.

Example 2:

Input: `usageLimits` $= [2, 1, 2]$
Output: 2
Explanation: In this example, we can use $0$ at most twice, $1$ at most once, and $2$ at most twice.
One way of creating the maximum number of groups while satisfying the conditions is:
Group 1 contains the number $[0]$.
Group 2 contains the numbers $[1, 2]$.
It can be shown that the maximum number of groups is 2.
So, the output is 2.

Example 3:

Input: `usageLimits` $= [1, 1]$
Output: 1
Explanation: In this example, we can use both $0$ and $1$ at most once.
One way of creating the maximum number of groups while satisfying the conditions is:
Group 1 contains the number $[0]$.
It can be shown that the maximum number of groups is 1.
So, the output is 1.

Constraints:

$1 \leq$ `usageLimits.length` $\leq 10^5$
$1 \leq$ `usageLimits[i]` $\leq 10^9$

For easier analysis, we let $n =$ `usageLimits.length` and $m = \max_i$ `usageLimits[i]`. ($m$ is not used in our analysis, but for naive greedy assignment algorithms, $m$ is an important factor.)

It turns out that there exists an efficient $O(n \log n)$ algorithm for this problem, assuming standard arithmetic operations can be performed in $O(1)$ time (i.e. RAM model). The algorithm involves binary searching for the correct output, performing a feasibility check at each iteration $k$ of the binary search.

The feasibility check involves checking the Gale-Ryser condition (Gale, 1957; Ryser, 1957). In the $t$-th iteration, we perform a weaker check to determine whether it is possible to assign indices to the $t$ largest groups (i.e. groups $k, k-1, \ldots, k-t+2, k-t+1$). This is done by checking the *demand*, or the number of elements in the $t$ largest groups (i.e. $\sum_{i=k-t+1}^{k} i = (k - (k-t+1) + 1)(k + (k-t+1))/2 = t(2k-t+1)/2$), against the *supply*, or the number of elements which can fit in $t$ groups if each group must have distinct elements (i.e. $\sum_{i=1}^{n} \min($`usageLimits[i]`$, t)$). The correctness of this feasibility check is given by the Gale-Ryser theorem (Gale, 1957; Ryser, 1957).

Notice that where $i^* = \min_{i:\texttt{usageLimit[i]}>t}$, $\texttt{supply}_t = \sum_{i=1}^n \min(\texttt{usageLimits[i]}, t) = \sum_{i=1}^{i^*-1} \texttt{usageLimits[i]} + \sum_{i=i^*}^n t = \sum_{i=1}^{i^*-1} \texttt{usageLimits[i]} + t(n - i^* + 1)$. $i^*$ can be found via binary search, so by pre-computing $\sum_{i=1}^{i^*-1} \texttt{usageLimits[i]}$, $\texttt{supply}_t$ can be computed in $\log n$ time. Thereafter, by pre-computing $\texttt{supply}_t$ for $t \in [1, n]$, the feasibility check can be done in $O(n)$ time, yielding an $O(n \log n)$ algorithm. (As it turns out, an $\tilde{O}(n)$ algorithm is sufficient to pass the time constraints for this problem, and so pre-computing $\texttt{supply}_t$ was not necessary to pass this problem.)

Initially, GPT o4-mini gets the binary search algorithm, but performs the feasibility check incorrectly. Instead of checking if $\forall t \in [1, k], \texttt{demand}_{k,t} \leq \texttt{supply}_t$, it simply checks if $\texttt{demand}_{k,k} \leq \texttt{supply}_k$:

```python
import sys

def main():
    data = sys.stdin.read().strip()
    # Input is given as a Python-style list, e.g. "[1, 2, 5]"
    usageLimits = eval(data)
    n = len(usageLimits)

    # Feasibility check: can we form k groups of sizes 1,2,...,k ?
    def canMake(k):
        # Sum of min(usageLimits[i], k) across all i
        s = 0
        # Early exit if sum already big enough
        target = k * (k + 1) // 2
        for x in usageLimits:
            s += x if x < k else k
            if s >= target:
                return True
        return s >= target

    # Binary-search k in [0..n]
    low, high = 0, n
    while low < high:
        mid = (low + high + 1) // 2
        if canMake(mid):
            low = mid
        else:
            high = mid - 1

    sys.stdout.write(str(low))

if __name__ == "__main__":
    main()
```

Surprisingly, this algorithm seems to work in most cases, even for the initially generated edge cases. For example, it returns the correct answer if `usageLimits` is an array consisting of repetitions of the same number (e.g. `[1, 1, 1, ..., 1, 1]`), or if it consists of only consecutive numbers (e.g. `[1, 2, 3, ..., 19, 20]`). As a result, the initial test set and coverage-guided tests fail to find bugs.

However, upon being asked to identify a counter-example, the LLM returns the test case `usageLimits = [4, 4, 1, 1]`. In the above code, `canMake(4)` returns **True** since $\texttt{demand}_{4,4} = \texttt{supply}_4 = 10$. However, we can easily notice that group 3 must contain at least one of the last 2 indices, whilst group 4 must contain both of the last 2 indices. Thus, one of the last two indices must be used twice, which is not possible. Indeed, we can verify that $\texttt{demand}_{4,3} = 9 > 8 = \texttt{supply}_3$.

After reflecting on the counter-example, GPT o4-mini comes up with a correct solution:

```python
import sys
import threading
def main():
    import sys, bisect
    data = sys.stdin.read().strip()
    usage = eval(data)
    n = len(usage)
    usage.sort()
    # build prefix sums
    prefix = [0]*(n+1)
    for i,u in enumerate(usage):
        prefix[i+1] = prefix[i] + u

    # check whether we can build k increasing groups 1,2,...,k
    def feasible(k):
        # for each t from 1..k ensure total_supply(t) >= demand(t)
        # total_supply(t) = sum_i min(usage[i], t)
        # demand(t) = sum of the largest t group-sizes = t*(2*k -
        #   t +1)//2
        for t in range(1, k+1):
            # how many usage[i] <= t
            idx = bisect.bisect_right(usage, t)
            supply = prefix[idx] + (n-idx)*t
            demand = t*(2*k - t + 1)//2
            if supply < demand:
                return False
        return True

    # binary search on k in [0..n]
    lo, hi = 0, n
    while lo < hi:
        mid = (lo + hi + 1)//2
        if feasible(mid):
            lo = mid
        else:
            hi = mid - 1

    # since groups start at size 1, answer = lo
    sys.stdout.write(str(lo))

if __name__ == "__main__":
    main()
```

## G    REGRESSION CASE STUDY

Here, we examine how ICE-Coder resulted in a regression for the "Sandwiches" problem (question ID abc318_e of LiveCodeBench taken from AtCoder) when run on DeepSeek V3.

The "Sandwiches" problem is reproduced here:

> You are given a sequence of positive integers of length $N$: $A = (A_1, A_2, \ldots, A_N)$. Find the number of triples of positive integers $(i, j, k)$ that satisfy all of the following conditions:
>
> - $1 \leq i < j < k \leq N$,
> - $A_i = A_k$,
> - $A_i \neq A_j$.

Input

The input is given from Standard Input in the following format:

$N$

$A_1 A_2 \ldots A_N$

Output

Print the answer as an integer.

Constraints

- $3 \leq N \leq 3 \times 10^5$
- $1 \leq A_i \leq N$
- All input values are integers.

Sample Input 1

5

1 2 1 3 2

Sample Output 1

3

The following three triples of positive integers $(i, j, k)$ satisfy the conditions:

- $(i, j, k) = (1, 2, 3)$
- $(i, j, k) = (2, 3, 5)$
- $(i, j, k) = (2, 4, 5)$

Sample Input 2

7

1 2 3 4 5 6 7

Sample Output 2

0

There may be no triples of positive integers $(i, j, k)$ that satisfy the conditions.

Sample Input 3

13

9 7 11 7 3 8 1 13 11 11 11 6 13

Sample Output 3

20

A correct $O(N)$ solution to the problem is to iterate through the list $A$ once, keeping track of the total number of valid triples in the subarray $A_1, A_2, \ldots, A_k$, $\texttt{runningTotal}_k$, in the $k$-th iteration. In iteration $k$, we add the number of new triples involving the $k$-th element of $A$ to $\texttt{runningTotal}$. Therefore, in iteration $k$, we have to find the number of triples involving the index $k$ in the subarray $A_1, A_2, \ldots, A_k$.

In order to do so, let us introduce some supplementary notation. Let $s$ be the integer at $A_k$. Let $\texttt{count}_k$ be the number of occurrences of $s$ in $A_1, \ldots, A_k$ (i.e. the index $k$ is the $\texttt{count}_k$-th occurrence of $s$ in the array $A$). We let $\texttt{other}_k = k - \texttt{count}_k$ be the number of integers in the subarray $A_1, A_2 \ldots, A_k$ which are not $s$. We overload this notation and let $\texttt{other}_{i,k} = \texttt{other}_k - \texttt{other}_i$ be the number of integers in the subarray $A_{i+1}, A_{i+2} \ldots, A_k$ which are not $s$. Then, we have:

$$\texttt{runningTotal}_k - \texttt{runningTotal}_{k-1}$$

$$= \sum_{i=1}^{k} \mathbb{1}_{A_i = A_k}(\texttt{other}_{i,k})$$

$$= \sum_{i=1}^{k} \mathbb{1}_{A_i = A_k}(\texttt{other}_k - \texttt{other}_i)$$

$$= \sum_{i=1}^{k} \mathbb{1}_{A_i = A_k}(\texttt{other}_k) - \sum_{i=1}^{k} \mathbb{1}_{A_i = A_k}(\texttt{other}_i)$$

$$= (\texttt{count}_k)(\texttt{other}_k) - \sum_{i=1}^{k} \mathbb{1}_{A_i = A_k}(\texttt{other}_i)$$

$$= (\texttt{count}_k)(k - \texttt{count}_k) - \sum_{i=1}^{k} \mathbb{1}_{A_i = A_k}(i - \texttt{count}_i)$$

It turns out that keeping track of $\texttt{count}_k$ and $\sum_{i=1}^{k} \mathbb{1}_{A_i = A_k}(i - \texttt{count}_i)$ using hash maps (or Python dictionaries) can be done efficiently in expectation.

DeepSeek V3's initial code correctly applies this method, solving the problem (albeit with some redundant steps):

```python
import sys
from collections import defaultdict

def solve():
    input = sys.stdin.read().split()
    ptr = 0
    N = int(input[ptr])
    ptr += 1
    A = list(map(int, input[ptr:ptr+N]))

    # Step 1: Compute prefix counts
    prefix = [0] * (N + 1)
    count = defaultdict(int)
    for k in range(1, N + 1):
        x = A[k-1]
        prefix[k] = count[x]
        count[x] += 1

    # Step 2: Reset count and compute running sums
    count = defaultdict(int)
    running_sum = defaultdict(int)
    answer = 0
    for k in range(1, N + 1):
        x = A[k-1]
        current_prefix = prefix[k]
        c = count[x]
        answer += c * (k - current_prefix) - running_sum[x]
        running_sum[x] += (k - current_prefix)
        count[x] += 1
    print(answer)
```

```
solve()
```

However, on the input:

```
7
3 3 3 5 3 3 3
```

the expected output in the test generated was 10, where the correct output should be 9 — there are $3 \times 3$ ways to pair up one index from the left of the 5, and one from the right of it.

Yet, in the diagnosis phase, the output 10 is picked due to a calculation error and a hallucinated triple:

```
To solve this problem, we need to find the number of triples (i, j
    , k) such that:
1. \(1 \leq i < j < k \leq N\),
2. \(A_i = A_k\),
3. \(A_i \neq A_j\).

### Step−by−Step Reasoning:

1. **Understanding the Problem**:
   − We need to count all possible triples (i, j, k) where the
       first and third elements are equal, and the middle element
       is different from them.
   − The indices must be in increasing order: \(i < j < k\).

2. **Key Insight**:
   − For each value \(x\) in the array, we can consider all pairs
       of indices (i, k) where \(A_i = A_k = x\) and \(i < k\).
   − For each such pair (i, k), the number of valid \(j\)'s is the
       number of elements between \(i\) and \(k\) (i.e., \(k − i
       − 1\)) minus the number of elements between \(i\) and \(k\)
       that are equal to \(x\).

3. **Efficient Calculation**:
   − For each occurrence of \(x\), we can keep track of their
       positions.
   − For each \(x\), iterate over all pairs of its positions (i, k
       ) where \(i < k\).
   − For each pair (i, k), the number of valid \(j\)'s is \((k − i
       − 1) − (\text{number of } x \text{'s between } i \text{
       and } k)\).
   − Summing this over all valid pairs (i, k) for all \(x\) gives
       the total count.

4. **Optimization**:
   − Instead of iterating over all pairs (i, k) for each \(x\), we
       can use prefix sums to compute the number of valid \(j\)'s
       efficiently.
   − For each \(x\), let the positions be \(p_1, p_2, \ldots, p_m
       \). The total valid \(j\)'s for all pairs (i, k) is:
       \[
       \sum_{i=1}^{m} \sum_{k=i+1}^{m} (p_k − p_i − 1) − (k − i − 1)
       \]
       Simplifying, this becomes:
       \[
       \sum_{i=1}^{m} \sum_{k=i+1}^{m} (p_k − p_i − 1 − (k − i − 1))
           = \sum_{i=1}^{m} \sum_{k=i+1}^{m} (p_k − p_i − k + i)
```

```
\]
Further simplification:
\[
\sum_{i=1}^{m} \sum_{k=i+1}^{m} (p_k - k) - (p_i - i)
\]
```
- This can be computed efficiently using prefix sums over \(p_k - k\) for each \(x\).

5. **Applying to the Given Input**:
   - Input:
     ```
     7
     3 3 3 5 3 3 3
     ```
   - The positions of 3 are: 1, 2, 3, 5, 6, 7 (0-based or 1-based? Assuming 1-based here).
   - For \(x = 3\), the positions are \(p = [1, 2, 3, 5, 6, 7]\).
   - Compute \(p_k - k\) for each \(p_k\):
     - \(p_1 - 1 = 0\),
     - \(p_2 - 2 = 0\),
     - \(p_3 - 3 = 0\),
     - \(p_4 - 4 = 1\),
     - \(p_5 - 5 = 1\),
     - \(p_6 - 6 = 1\).
   - The prefix sum array \(S\) for \(p_k - k\) is \([0, 0, 0, 1, 2, 3]\).
   - For each \(i\), the sum of \((p_k - k) - (p_i - i)\) for \(k > i\) is:
     - For \(i=1\): \((0 + 0 + 1 + 2 + 3) - 5 \cdot 0 = 6\),
     - For \(i=2\): \((0 + 1 + 2 + 3) - 4 \cdot 0 = 6\),
     - For \(i=3\): \((1 + 2 + 3) - 3 \cdot 0 = 6\),
     - For \(i=4\): \((1 + 1) - 2 \cdot 1 = 0\),
     - For \(i=5\): \(1 - 1 \cdot 1 = 0\),
     - For \(i=6\): \(0\).
   - Total sum: \(6 + 6 + 6 + 0 + 0 + 0 = 18\).
   - However, this seems incorrect because the expected answer is likely smaller. Let's re-examine the calculation.

6. **Alternative Approach**:
   - For each pair (i, k) where \(A_i = A_k = 3\), count the number of \(j\)'s between \(i\) and \(k\) where \(A_j \neq 3\).
   - Pairs of (i, k):
     - (1, 2): \(j\)'s: none (since \(k - i - 1 = 0\)),
     - (1, 3): \(j = 2\) (valid, since \(A_2 = 3 \neq 3\)? No, \(A_2 = 3\), so invalid),
     - (1, 5): \(j = 2, 3, 4\) (valid \(j\)'s: 4, since \(A_4 = 5 \neq 3\)),
     - (1, 6): \(j = 2, 3, 4, 5\) (valid \(j\)'s: 4),
     - (1, 7): \(j = 2, 3, 4, 5, 6\) (valid \(j\)'s: 4),
     - (2, 3): \(j\)'s: none,
     - (2, 5): \(j = 3, 4\) (valid \(j\)'s: 4),
     - (2, 6): \(j = 3, 4, 5\) (valid \(j\)'s: 4),
     - (2, 7): \(j = 3, 4, 5, 6\) (valid \(j\)'s: 4),
     - (3, 5): \(j = 4\) (valid \(j\)'s: 4),
     - (3, 6): \(j = 4, 5\) (valid \(j\)'s: 4),
     - (3, 7): \(j = 4, 5, 6\) (valid \(j\)'s: 4),
     - (5, 6): \(j\)'s: none,

- (5, 7): $\(j = 6\)$ (valid $\(j\)$'s: 6, since $\(A_6 = 3 \neq$
  $3\)$? No, invalid),
- (6, 7): $\(j\)$'s: none.
- Summing the valid $\(j\)$'s:
- (1, 5): 1,
- (1, 6): 1,
- (1, 7): 1,
- (2, 5): 1,
- (2, 6): 1,
- (2, 7): 1,
- (3, 5): 1,
- (3, 6): 1,
- (3, 7): 1,
- (5, 7): 1.
- Total: 10 valid triples.

### Final Answer:
The correct output is `10`.

ANSWER:
```
10
```

When modifying the code, DeepSeek V3 again miscounts the number of triples, this time undercounting. However, it trusts our expected output. In any case, it produces an incorrect piece of code, leading to a regression.

### Understanding the Problem

The problem requires counting the number of triples (i, j, k) in a
    sequence of integers where:
1. $\(1 \leq i < j < k \leq N\)$,
2. $\(A_i = A_k\)$,
3. $\(A_i \neq A_j\)$.

Given the constraints (N up to $\(3 \times 10^5\)$), the solution
    must be efficient, likely $\(O(N)\)$ or $\(O(N \log N)\)$.

### Analyzing the Given Code

The provided code attempts to solve the problem in two steps:
1. **Prefix Count Calculation**: For each position $\(k\)$, it
    calculates how many times $\(A_k\)$ has appeared before $\(k\)$ (
    stored in `prefix[k]`).
2. **Running Sum Calculation**: For each $\(k\)$, it maintains a
    running sum of certain values to compute the answer.

The core logic for counting valid triples is in the second loop (
    lines 23–29). The formula used is:
$\[ \text{answer} += c \times (k - \text{current\_prefix}) - \text{
    running\_sum}[x] \]$
where:
- $\(c\)$ is the count of how many times $\(x = A_k\)$ has been seen
    so far.
- $\(\text{current\_prefix}\)$ is the number of times $\(x\)$ appeared
    before $\(k\)$ (from the first step).
- $\(\text{running\_sum}[x]\)$ is a cumulative value for $\(x\)$.

### Identifying the Bug

The code fails on the input:
```
7
3 3 3 5 3 3 3
```
Expected output: 10, but the code outputs 9.

Let's manually compute the expected triples:
- For $i=1$ (A=3), possible $k$ values are 3,5,6,7 (since $A_i = A_k = 3$)).
  - For $k=3$: $j$ can be 2 (A=3, but $A_i = A_j = 3$), invalid).
  - For $k=5$: $j$ can be 2,3,4 (A=3,3,5). Valid $j$ is 4 (A=5).
  - For $k=6$: $j$ can be 2,3,4,5 (A=3,3,5,3). Valid $j$ are 4 (A=5).
  - For $k=7$: $j$ can be 2,3,4,5,6 (A=3,3,5,3,3). Valid $j$ are 4 (A=5).
  - Total for $i=1$: 1 (k=5) + 1 (k=6) + 1 (k=7) = 3.

- For $i=2$ (A=3), possible $k$ values are 5,6,7.
  - For $k=5$: $j$ can be 3,4 (A=3,5). Valid $j$ is 4 (A=5).
  - For $k=6$: $j$ can be 3,4,5 (A=3,5,3). Valid $j$ is 4 (A=5).
  - For $k=7$: $j$ can be 3,4,5,6 (A=3,5,3,3). Valid $j$ is 4 (A=5).
  - Total for $i=2$: 1 + 1 + 1 = 3.

- For $i=3$ (A=3), possible $k$ values are 6,7.
  - For $k=6$: $j$ can be 4,5 (A=5,3). Valid $j$ is 4 (A=5).
  - For $k=7$: $j$ can be 4,5,6 (A=5,3,3). Valid $j$ is 4 (A=5).
  - Total for $i=3$: 1 + 1 = 2.

- For $i=4$ (A=5), no $k$ exists where $A_k = 5$ and $k > 4$ (since $A_4=5$ and next 5 is beyond the array).
  - Total for $i=4$: 0.

- For $i=5$ (A=3), possible $k$ is 7.
  - For $k=7$: $j$ can be 6 (A=3). Invalid since $A_i = A_j = 3$).
  - Total for $i=5$: 0.

- For $i=6$ (A=3), no $k$ exists beyond it.
  - Total for $i=6$: 0.

Summing up: 3 (i=1) + 3 (i=2) + 2 (i=3) = 8. Wait, this contradicts the expected output of 10. Hmm, perhaps I missed some cases. Let me recount:

Wait, for $i=1, k=5$: $j$ can be 2,3,4 (A=3,3,5). Valid $j$ is 4 (A=5). So 1.
For $i=1, k=6$: $j$ can be 2,3,4,5 (A=3,3,5,3). Valid $j$ is 4 (A=5). So 1.
For $i=1, k=7$: $j$ can be 2,3,4,5,6 (A=3,3,5,3,3). Valid $j$ is 4 (A=5). So 1.
Total for $i=1$: 3.

For $(i=2, k=5)$: $(j)$ can be 3,4 (A=3,5). Valid $(j)$ is 4 (A=5). So 1.
For $(i=2, k=6)$: $(j)$ can be 3,4,5 (A=3,5,3). Valid $(j)$ is 4 (A=5). So 1.
For $(i=2, k=7)$: $(j)$ can be 3,4,5,6 (A=3,5,3,3). Valid $(j)$ is 4 (A=5). So 1.
Total for $(i=2)$: 3.

For $(i=3, k=6)$: $(j)$ can be 4,5 (A=5,3). Valid $(j)$ is 4 (A=5). So 1.
For $(i=3, k=7)$: $(j)$ can be 4,5,6 (A=5,3,3). Valid $(j)$ is 4 (A=5). So 1.
Total for $(i=3)$: 2.

For $(i=4, k=...)$: No $(k)$ with $(A_k=5)$ beyond $(i=4)$. So 0.

For $(i=5, k=7)$: $(j)$ can be 6 (A=3). Invalid since $(A_i = A_j = 3)$. So 0.

Total: 3 + 3 + 2 = 8. But expected is 10. Maybe the expected output is wrong? Or perhaps the counting is missing something.

Wait, perhaps the expected output is incorrect, but the code outputs 9, which is also not matching our manual count of 8. This suggests the code's logic might be overcounting or undercounting in some way.

### Debugging the Code

The code's logic is non-trivial. Let's try to understand what it's doing:
1. `prefix[k]` is the number of times $(A_k)$ has appeared before $(k)$.
2. In the second loop, for each $(k)$, it updates `answer` using:
$$ \text{answer} += c \times (k - \text{current\_prefix}) - \text{running\_sum}[x] $$
where:
   - $(c)$ is the count of $(x)$ so far.
   - $(k - \text{current\_prefix})$ is roughly the number of $(j)$'s where $(A_j \neq x)$ between the last occurrence of $(x)$ and $(k)$.
   - `running_sum[x]` seems to accumulate these values.

The issue might be that the formula doesn't correctly account for all valid $(j)$'s when multiple $(i)$'s are involved. Specifically, the subtraction of `running_sum[x]` might be causing undercounting.

### Correct Approach

A better approach is to:
1. For each value $(x)$, maintain a list of its indices.
2. For each occurrence of $(x)$ at position $(k)$, the number of valid $(i)$'s is the number of $(i)$'s before $(k)$ where $(A_i = x)$.
3. For each such $(i)$, the number of valid $(j)$'s is the number of elements between $(i)$ and $(k)$ that are not $(x)$.

This can be optimized using prefix sums:
- Precompute prefix sums where at each position, we count how many times each value has occurred up to that point.
- For each $x$, iterate over its indices, and for each pair $(i, k)$, the number of valid $j$'s is $(k - i - 1) - (\text{prefix}[k-1][x] - \text{prefix}[i][x])$.

However, with $N$ up to $3 \times 10^5$, we need an $O(N)$ solution. The given code attempts this but seems to have a flaw in the formula.

### Suggested Fix

The bug is in the formula:
$$ \text{answer} += c \times (k - \text{current\_prefix}) - \text{running\_sum}[x] $$
This seems to undercount because it subtracts `running_sum[x]` which might not correctly account for all valid $j$'s.

Instead, we can:
1. For each $x$, keep track of the sum of indices where $x$ has appeared.
2. For each new occurrence of $x$ at $k$, the number of new valid triples is:
$$ (\text{count}[x] \times (k - 1) - \text{sum\_indices}[x]) - \text{prefix}[k] $$
where:
- $\text{count}[x]$ is the number of times $x$ has appeared before $k$.
- $\text{sum\_indices}[x]$ is the sum of indices where $x$ has appeared before $k$.
- $\text{prefix}[k]$ is the number of times $x$ has appeared before $k$ (which seems redundant here).

This is complex, and perhaps a simpler approach is to iterate over all possible $x$, and for each $x$, iterate over its indices, and for each pair of indices $(i, k)$, the number of valid $j$'s is the number of elements between $i$ and $k$ that are not $x$. This can be optimized using prefix sums.

### Rewriting the Code

Here's a corrected approach:
1. Precompute for each position, the number of times each value has occurred up to that point (prefix counts).
2. For each value $x$, iterate over its indices, and for each pair $(i, k)$, the number of valid $j$'s is:
$$ (k - i - 1) - (\text{prefix}[k-1][x] - \text{prefix}[i][x]) $$
which simplifies to:
$$ (k - i - 1) - (\text{number of } x \text{'s between } i \text{ and } k) $$

However, with $N$ up to $3 \times 10^5$, we need to avoid $O(N^2)$ operations. The given code's approach is likely correct in spirit but has an off-by-one or similar error.

### Final Answer

The bug in the code is in the formula used to update `answer` in the second loop. The current formula undercounts because it incorrectly subtracts `running_sum[x]`. A corrected version should more carefully account for the valid $j$'s between $i$ and $k$.

Here's a corrected version of the code:

```python
import sys
from collections import defaultdict

def solve():
    input = sys.stdin.read().split()
    ptr = 0
    N = int(input[ptr])
    ptr += 1
    A = list(map(int, input[ptr:ptr+N]))

    # Step 1: Compute prefix counts
    prefix = [0] * (N + 1)
    count = defaultdict(int)
    for k in range(1, N + 1):
        x = A[k-1]
        prefix[k] = count[x]
        count[x] += 1

    # Step 2: Reset count and compute running sums
    count = defaultdict(int)
    running_sum = defaultdict(int)
    answer = 0
    for k in range(1, N + 1):
        x = A[k-1]
        current_prefix = prefix[k]
        c = count[x]
        answer += c * (k - 1 - current_prefix) - running_sum[x]
        running_sum[x] += (k - 1 - current_prefix)
        count[x] += 1
    print(answer)

solve()
```

Key changes:
1. Changed `c * (k - current_prefix)` to `c * (k - 1 - current_prefix)` to correctly account for the indices.
2. This adjustment ensures that the number of valid $j$'s is correctly calculated between $i$ and $k$.

This should now correctly output `10` for the input:
```
7
3 3 3 5 3 3 3
```

# H PROMPTS

We detail the prompts used in this section. The prompts were adapted from and inspired by those used in AgentCoder Huang et al. (2024b). The parts highlighted in **RED** are to be substituted with appropriate values for the problem.

## H.1 CODE GENERATION

**Initial Generation**

```
**Task**: As a competitive programmer familiar with
    advanced data structures and algorithms, you are
    required to solve a given coding problem. Use a
    Chain-of-Thought approach to break down the
    problem, find a correct and efficient algorithm,
    create pseudocode, and then write the code in
    Python language.

**Instructions**:
1. **Understand and Clarify**: Make sure you
    understand the task.
2. **Brainstorm**: Relate the task to similar problems
    , and identify any relevant data structures and
    algorithms.
3. **Analyze**: Using examples, edge cases, and
    extreme cases, analyze various methods to solve
    the problem, with a focus on correctness and
    efficiency.
4. **Time Complexity**: Analyze the time complexity of
     the algorithms.
5. **Algorithm/Method Selection**: Decide on the most
    efficient way to solve the task correctly.
6. **Pseudocode Creation**: Write down the steps you
     will follow in pseudocode.
7. **Code Generation**: Translate your pseudocode into
     annotated, executable Python code, leaving
    comments explaining each step in the code.

**Code Formatting**: Please write code in
```python
[REPLACE THIS WITH YOUR CODE]
```
format.

# For example:

## Example Prompt:
Output the sum of a list of numbers.
The first line is the length of the list, the next
    line is the space-delimited list of numbers.

## Example Completion:
```python
import sys
n = eval(sys.stdin.readline().rstrip('\r\n'))
lst = [int(i) for i in sys.stdin.readline().rstrip('\r
    \n').split()]
output = 0
for i in range(n):
```
```

```
        output += lst[i]
    sys.stdout.write(f'{output}\n')
    ```
```

**Problem**:

**PROBLEM_DESCRIPTION**

Use a Chain-of-Thought approach to break down the
    problem, find a correct and efficient algorithm,
    create pseudocode, and then write a correct code
    in Python language.

**Instructions**:
1. **Understand and Clarify**: Make sure you
    understand the task.
2. **Brainstorm**: Relate the task to similar problems
    , and identify any relevant data structures and
    algorithms.
3. **Analyze**: Using examples, edge cases, and
    extreme cases, analyze various methods to solve
    the problem, with a focus on correctness and
    efficiency.
4. **Time Complexity**: Analyze the time complexity of
     the algorithms.
5. **Algorithm/Method Selection**: Decide on the most
    efficient way to solve the task correctly.
6. **Pseudocode Creation**: Write down the steps you
    will follow in pseudocode.
7. **Code Generation**: Translate your pseudocode into
     annotated, executable Python code, leaving
    comments explaining each step in the code.

In the time complexity analysis, keep in mind that
    operations on big integers in Python are not
    always constant-time. Optimize accordingly.

**Code Formatting**: Please write code in
```python
[REPLACE THIS WITH YOUR CODE]
```
format.

Remember to read the input from STDIN using `sys.stdin
    .read` and write the output to STDOUT using `sys.
    stdin.write`.
Include all necessary import statements but WRITE ONLY
    THE SOLUTION AND DO NOT PROVIDE TEST CASES.

**Reflection**

**Task**: As a programmer, you are required to find
    the bug in a code. Use a Chain-of-Thought approach
     to trace the code and identify the bug.

**Instructions**:

1. \*\*Understand and Clarify\*\*: Make sure you
   understand the problem.
2. \*\*Code Tracing\*\*: Understand how the code runs.
3. \*\*Reflection\*\*: Reflect on whether each execution
   state is correct.
4. \*\*Bug Identification\*\*: Identify the problem with
   the code.

**PROBLEM_DESCRIPTION**

\*\*Code\*\*:
**GENERATED_CODE**

\*\*Issue\*\*:

**TEST_INPUTS**

The code is expected to **EXPECTED_OUTPUTS**

But the code **ERRONEOUS_BEHAVIOUR**.

Below is a snippet of the execution trace of the code
    on the inputs:
**EXECUTION_TRACE**

Identify the problem with the code. Describe in detail
    why the code went wrong.
Suggest if the code should be modified or re-written
    entirely.

**Code Refinement**

\*\*Task\*\*: As a competitive programmer familiar with
    advanced data structures and algorithms, you are
    required to solve a given coding problem. Use a
    Chain-of-Thought approach to break down the
    problem, find a correct and efficient algorithm,
    create pseudocode, and then write the code in
    Python language.

\*\*Instructions\*\*:
1. \*\*Understand and Clarify\*\*: Make sure you
   understand the task.
2. \*\*Brainstorm\*\*: Relate the task to similar problems
   , and identify any relevant data structures and
   algorithms.
3. \*\*Analyze\*\*: Using examples, edge cases, and
   extreme cases, analyze various methods to solve
   the problem, with a focus on correctness and
   efficiency.
4. \*\*Time Complexity\*\*: Analyze the time complexity of
    the algorithms.
5. \*\*Algorithm/Method Selection\*\*: Decide on the most
   efficient way to solve the task correctly.
6. \*\*Pseudocode Creation\*\*: Write down the steps you
   will follow in pseudocode.

7. **Code Generation**: Translate your pseudocode into
   annotated, executable Python code, leaving
   comments explaining each step in the code.

**Code Formatting**: Please write code in
```python
[REPLACE THIS WITH YOUR CODE]
```
format.

# For example:

## Example Prompt:
Output the sum of a list of numbers.
The first line is the length of the list, the next
    line is the space-delimited list of numbers.

## Example Completion:
```python
import sys
n = eval(sys.stdin.readline().rstrip('\r\n'))
lst = [int(i) for i in sys.stdin.readline().rstrip('\r
    \n').split()]
output = 0
for i in range(n):
    output += lst[i]
sys.stdout.write(f'{output}\n')
```

**Problem**:

**PROBLEM_DESCRIPTION**

**Existing Incorrect Code**:
The following code was tried on the problem.
**GENERATED_CODE**

However, the code fails on:
**TEST_INPUTS**

The existing code does not produce the expected output
    on the above inputs.

A possible reason that the code provided does not work
    is that:
**REFLECTIONS**

You may choose to modify the existing code, or to come
    up with an entirely new code and algorithm. Pick
    whichever approach is more appropriate.
Use a Chain-of-Thought approach to break down the
    problem, find a correct and efficient algorithm,
    create pseudocode, and then write a correct code
    in Python language.

**Instructions**:

1. **Understand and Clarify**: Make sure you understand the task.
2. **Brainstorm**: Relate the task to similar problems, and identify any relevant data structures and algorithms.
3. **Analyze**: Using examples, edge cases, and extreme cases, analyze various methods to solve the problem, with a focus on correctness and efficiency.
4. **Time Complexity**: Analyze the time complexity of the algorithms.
5. **Algorithm/Method Selection**: Decide on the most efficient way to solve the task correctly.
6. **Pseudocode Creation**: Write down the steps you will follow in pseudocode.
7. **Code Generation**: Translate your pseudocode into annotated, executable Python code, leaving comments explaining each step in the code.

In the time complexity analysis, keep in mind that operations on big integers in Python are not always constant-time. Optimize accordingly.

**Code Formatting**: Please write code in
```python
[REPLACE THIS WITH YOUR CODE]
```
format.

Remember to read the input from STDIN using `sys.stdin.read` and write the output to STDOUT using `sys.stdin.write`.
Include all necessary import statements but WRITE ONLY THE SOLUTION AND DO NOT PROVIDE TEST CASES.

## H.2 TEST GENERATION

**Basic Test Cases**

**Role**: As a tester, your task is to create comprehensive test cases for a given problem.

**Instructions**:
- Implement an extensive and comprehensive set of test cases which conform to the problem constraints.

The format of each test case should be:
Inputs:
```

Input line 1
Input line 2
Input line 3
...
```

Outputs:
```

Output line 1
Output line 2

```
Output line 3
...
```

For large test cases, you may choose to provide a
    python expression instead. If you choose to
    provide a python expression, label the code block
    appropriately as in:
```python
REPLACE THIS LINE WITH THE PYTHON EXPRESSION
```

Provide either the plaintext or the python expression,
    NOT both.

# For example:

## Example Prompt:
Given list of numbers, determine if any two numbers
    are closer to each other than given threshold.
The first line of the input is the length of the list
    n, and the threshold t.
The next n lines of the input are the numbers in the
    list.
The output is a single line. which is True if any two
    numbers in the list are closer than the threshold.

## Example Completion:
Inputs:
```
3 1
1.0
2.0
3
```

Outputs:
```
False
```

Inputs:
```python
"10000 0.5\n" + " ".join(range(10000))
```

Outputs:
```
False
```

**Problem**:

PROBLEM_DESCRIPTION

Remember to follow the format described and ENCLOSE
    THE TEST INPUTS AND EXPECTED OUTPUTS IN CODE
    BLOCKS as in:
Inputs:
```

```
Input line 1
Input line 2
Input line 3
...
```

Outputs:
```
Output line 1
Output line 2
Output line 3
...
```

If you choose to provide a python expression instead,
    label the code block appropriately as in:
```python
REPLACE THIS LINE WITH THE PYTHON EXPRESSION
```

DO NOT provide both the plaintext and python
    expression for the same test case.
Only the test case inputs/outputs should be in the
    code blocks. DO NOT WRITE COMMENTS IN THE CODE
    BLOCKS.
Ensure that the generated tests conform to the input
    constraints and format.
Write the test cases in FULL and do not shorten them
    with ellipses.
DO NOT GENERATE A SOLUTION TO THE PROBLEM.

**Edge Scenarios**

**Role**: As a tester, your task is to think of a
    comprehensive list of edge scenarios for a given
    problem. These edge scenarios should test the code
    's robustness and reliability under extreme or
    unusual conditions.

**Instructions**:
- Come up with an extensive and comprehensive list of
    edge scenarios for the given problem.
- Pay special attention to edge cases as they often
    reveal hidden bugs.

The format of edge cases should be:
- Edge Case 1
- Edge Case 2
- Edge Case 3

# For example:

## Prompt 1:
Given list of numbers, determine if any two numbers
    are closer to each other than given threshold.

## Completion 1:
```

    - No two numbers are closer to each other than the
        threshold
    - Only one pair of numbers are closer to each other
        than the threshold
    - There are negative numbers
    - The difference between each pair of numbers is
        exactly the threshold
    - The threshold is 0
    - The threshold is infinite

    ## Prompt 2:
    Given a string containing multiple groups of nested
        parentheses, your goal is to separate each group
        with a space.
    Separate groups are balanced (each open brace is
        properly closed) and not nested within each other.

    ## Completion 2:
    - There is a single large group of nested parentheses
    - There are many small groups of nested parentheses
    - There are many layers of nesting of parentheses

    **Problem**:

    **PROBLEM DESCRIPTION**

    The edge scenarios should be unique but still able to
        conform to the input format and constraints.
    That is, the generated edge scenario should not be "
        ensure that invalid inputs are flagged", or
        anything similar.

    Remember to format the edge cases as separate bullet
        points with "-" like:
    - Edge Case 1
    - Edge Case 2
    - Edge Case 3

**Edge Test Cases**

    **Role**: As a tester, your task is to create
        comprehensive test cases for a given scenario in a
        given problem.

    **Instructions**:
    - Implement an extensive and comprehensive set of test
        cases which conform to the problem constraints.
    - Pay special attention to edge cases as they often
        reveal hidden bugs.

    The format of each test case should be:
    Inputs:
    ```
    Input line 1
    Input line 2
    Input line 3

```
...
```
Outputs:
```
Output line 1
Output line 2
Output line 3
...
```

For large test cases, you may choose to provide a
    python expression instead. If you choose to
    provide a python expression, label the code block
    appropriately as in:
```python
REPLACE THIS LINE WITH THE PYTHON EXPRESSION
```

Provide either the plaintext or the python expression,
    NOT both.

# For example:

## Example Prompt:
Given list of numbers, determine if any two numbers
    are closer to each other than given threshold.
The first line of the input is the length of the list
    n, and the threshold t.
The next n lines of the input are the numbers in the
    list.
The output is a single line. which is True if any two
    numbers in the list are closer than the threshold.

Test for scenarios where The difference between each
    pair of numbers is exactly the threshold

## Example Completion:
Inputs:
```
3 0.5
1.5
2
2.5
```
Outputs:
```
False
```

Inputs:
```
6 0.3
1.3
2.5
1.9
2.8
2.2
1.6
```
```

```
Outputs:
```
```
False
```

Inputs:
```python
"10000 1\n" + " ".join(range(10000))
```
Outputs:
```
False
```

**Problem**:

**PROBLEM DESCRIPTION**

Specifically, create a list of comprehensive tests for
    scenarios where **EDGE SCENARIO**

Remember to follow the format described and ENCLOSE
    THE TEST INPUTS AND EXPECTED OUTPUTS IN CODE
    BLOCKS as in:
Inputs:
```
Input line 1
Input line 2
Input line 3
...
```
Outputs:
```
Output line 1
Output line 2
Output line 3
...
```

If you choose to provide a python expression instead,
    label the code block appropriately as in:
```python
REPLACE THIS LINE WITH THE PYTHON EXPRESSION
```

DO NOT provide both the plaintext and python
    expression for the same test case.
Only the test case inputs/outputs should be in the
    code blocks. DO NOT WRITE COMMENTS IN THE CODE
    BLOCKS.
Ensure that the generated tests conform to the input
    constraints and format.
Write the test cases in FULL and do not shorten them
    with ellipses.
DO NOT GENERATE A SOLUTION TO THE PROBLEM.
```

**Large Test Cases**

```
**Role**: As a tester, your task is to create
    comprehensive test cases for a given problem.
    These test cases should encompass Large-Scale
    scenarios to ensure the code's performance,
    reliability, and scalability with large data
    samples.

**Instructions**:
- Implement a comprehensive set of test cases which
    conform to the problem constraints.
- Ensure each test case is well-documented with
    comments explaining the scenario it covers.
- Pay special attention to edge cases as they often
    reveal hidden bugs.
- For large-scale tests, focus on the function's
    efficiency and performance under heavy loads.

The format of each test case should be:
Inputs:
```

Input line 1
Input line 2
Input line 3

...
```

For large test cases, you may choose to provide a
    python expression instead. If you choose to
    provide a python expression, label the code block
    appropriately as in:
```python
REPLACE THIS LINE WITH THE PYTHON EXPRESSION
```

Provide either the plaintext or the python expression,
    NOT both.

You do not have to generate expected outputs.

# For example:

## Example Prompt:
Given list of numbers, determine if any two numbers
    are closer to each other than given threshold.
The first line of the input is the length of the list
    n, and the threshold t.
The next n lines of the input are the numbers in the
    list.
The output is a single line. which is True if any two
    numbers in the list are closer than the threshold.

## Example Completion:
Inputs:
```

3 1
1.0
2.0
3
```
```

```
Inputs:
```python
"10000 0.5\n" + " ".join(range(10000))
```
```

**Problem**:

**PROBLEM DESCRIPTION**

```
Remember to follow the format described and ENCLOSE
    THE TEST INPUTS IN CODE BLOCKS as in:
Inputs:
```
Input line 1
Input line 2
Input line 3
...
```

Outputs:
```
Output line 1
Output line 2
Output line 3
...
```

If you choose to provide a python expression instead,
    label the code block appropriately as in:
```python
REPLACE THIS LINE WITH THE PYTHON EXPRESSION
```

DO NOT provide both the plaintext and python
    expression for the same test case.
Only the test case inputs should be in the code blocks
    . DO NOT WRITE COMMENTS IN THE CODE BLOCKS.
Ensure that the generated tests conform to the input
    constraints and format.
GENERATE ONLY THE LARGE TEST INPUTS AND NOT THE
    EXPECTED OUTPUTS.
Write the test cases in FULL and DO NOT SHORTEN THEM
    WITH ELLIPSES.
DO NOT GENERATE A SOLUTION TO THE PROBLEM.
```

**Coverage-guided Test Cases**

```
**Role**: As a tester, your task is to create a test
    case for a piece of code for a given problem.

**Instructions**:

The format of each test case should be:
Inputs:
```
Input line 1
Input line 2
```

```
Input line 3
...
```
Outputs:
```
Output line 1
Output line 2
Output line 3
...
```

For large test cases, you may choose to provide a
    python expression instead. If you choose to
    provide a python expression, label the code block
    appropriately as in:
```python
REPLACE THIS LINE WITH THE PYTHON EXPRESSION
```

Provide either the plaintext or the python expression,
    NOT both.

# For example:

## Example Prompt:
Given list of numbers, determine if any two numbers
    are closer to each other than given threshold.
The first line of the input is the length of the list
    n, and the threshold t.
The next n lines of the input are the numbers in the
    list.
The output is a single line. which is True if any two
    numbers in the list are closer than the threshold.

## Example Completion:
Inputs:
```
3 1
1.0
2.0
3
```
Outputs:
```

False
```

Inputs:
```python
"10000 0.5\n" + " ".join(range(10000))
```
Outputs:
```

False
```

**Problem**:
```

**PROBLEM_DESCRIPTION**

\*\*Code\*\*:
**GENERATED_CODE**

Write a test case such that line **LINE_NUMBER** of the
    above code will be executed.
Use a Chain-of-Thought approach to break down the code
    and formulate a test case such that line
    **LINE_NUMBER** will be executed.

Remember to follow the format described and ENCLOSE
    THE TEST INPUTS AND EXPECTED OUTPUTS IN CODE
    BLOCKS as in:
Inputs:
```

Input line 1
Input line 2
Input line 3
...
```

Outputs:
```

Output line 1
Output line 2
Output line 3
...
```

If you choose to provide a python expression instead,
    label the code block appropriately as in:
```python
REPLACE THIS LINE WITH THE PYTHON EXPRESSION
```

DO NOT provide both the plaintext and python
    expression for the same test case.
Only the test case inputs/outputs should be in the
    code blocks. DO NOT WRITE COMMENTS IN THE CODE
    BLOCKS.
Ensure that the generated tests conform to the input
    constraints and format.
Write the test cases in FULL and do not shorten them
    with ellipses.
DO NOT GENERATE A SOLUTION TO THE PROBLEM.

**Coverage-guided Test Cases**

\*\*Role\*\*: As a tester, your task is to create a test
    case to expose a bug in a piece of code for a
    given problem.

\*\*Instructions\*\*:

The format of each test case should be:
Inputs:
```

```
   Input line 1
   Input line 2
   Input line 3
   ...
   ```
   Outputs:
   ```

   Output line 1
   Output line 2
   Output line 3
   ...
   ```

   For large test cases, you may choose to provide a
       python expression instead. If you choose to
       provide a python expression, label the code block
       appropriately as in:
   ```python
   REPLACE THIS LINE WITH THE PYTHON EXPRESSION
   ```

   Provide either the plaintext or the python expression,
       NOT both.

   # For example:

   ## Example Prompt:
   Given list of numbers, determine if any two numbers
       closer to each other than given threshold.
   The first line of the input is the length of the list
       n, and the threshold t.
   The next n lines of the input are the numbers in the
       list.
   The output is a single line. which is True if any two
       numbers in the list are closer than the threshold.

   ## Example Completion:
   Inputs:
   ```

   3 1
   1.0
   2.0
   3 ```

   Outputs:
   ```

   False
   ```

   Inputs:
   ```python
   "10000 0.5\n" + " ".join(range(10000))
   ```
   Outputs:
   ```

   False
   ```
```

\*\*Problem\*\*:

**PROBLEM_DESCRIPTION**

\*\*Existing Code\*\*:
**GENERATED_CODE**

It appears that the existing code is incorrect. Find a
    valid, constraint-adhering test case for which
    the existing code fails to output the correct
    answer.

Remember to follow the format described and ENCLOSE
    THE TEST INPUTS AND EXPECTED OUTPUTS IN CODE
    BLOCKS as in:
Inputs:
```
Input line 1
Input line 2
Input line 3
...
```

Outputs:
```
Output line 1
Output line 2
Output line 3
...
```

If you choose to provide a python expression instead,
    label the code block appropriately as in:
```python
REPLACE THIS LINE WITH THE PYTHON EXPRESSION
```

DO NOT provide both the plaintext and python
    expression for the same test case.
Only the test case inputs/outputs should be in the
    code blocks. DO NOT WRITE COMMENTS IN THE CODE
    BLOCKS.
Ensure that the generated tests conform to the input
    constraints and format. The code is NOT REQUIRED
    to handle inputs that violate the input
    constraints and format.
Write the test cases in FULL and do not shorten them
    with ellipses.
DO NOT GENERATE A SOLUTION TO THE PROBLEM.

## H.3 ERROR DIAGNOSIS

**Input Validation**

\*\*Task\*\*: As a tester, you are required to figure out
    why a code runs into an error on a given input.

Determine if the input is valid. Use a Chain−of−
Thought approach to identify the issue.

**Instructions**:
1. **Understand and Clarify**: Make sure you
   understand the problem.
2. **Input Parsing**: Interpret what the input means.
3. **Format Verification**: Compare the format of the
   input to that of the examples.
4. **Parameter Verification**: Ensure that the input
   is self−consistent.
5. **Logical Verification**: Ensure that the input has
    a corresponds to a valid problem instance with a
   valid solution.
6. **Answer**: Answer YES if the input is valid, else
   NO

**Answer Formatting**: Please write your answer on a
    single line after "ANSWER: " as
ANSWER: YES
or
ANSWER: NO

# For example:

## Prompt 1:
Output the sum of a list of numbers.
The first line is the length of the list, the next
    line is the space−delimited list of numbers.
Example:
```
4
1 2 3 4
```

Actual Input:
```
3
1 2
```

## Response 1:
ANSWER: NO

Reason: There are only 2 items in the list, not 3

## Prompt 2:
Output the sum of a list of numbers.
The first line is the length of the list, the next
    line is the space−delimited list of numbers.
Example:
```
4
"1" "2" "3" "4"
```

Input:
```
2
```

```
"1" "2"
```

## Response 2:
ANSWER: YES

Reason: The input format has quotation marks like in the example; there are two numbers in the list as stated in the paramters

**Problem**:

**PROBLEM_DESCRIPTION**

**Actual Input**:

**TEST_INPUTS**

**Instructions**:
1. **Understand and Clarify**: Make sure you understand the problem.
2. **Input Parsing**: Interpret what the input means.
3. **Format Verification**: Compare the format of the input to that of the examples.
4. **Parameter Verification**: Ensure that the input is self-consistent.
5. **Logical Verification**: Ensure that the input has a corresponds to a valid problem instance with a valid solution.
6. **Answer**: Answer YES if the input is valid, else NO

The input is provided as is to stdin (if there are ellipses and descriptions in the input, they are not expanded and the input thus is invalid).
Apart from trailing whitespaces, input format should match the examples given EXACTLY (if there are spaces/brackets/quotation marks in the format of the examples, the input should have them too).
Remember to write your answer on a single line after "ANSWER: " as
ANSWER: YES
or
ANSWER: NO

**Correct Output Judgement**

**Role**: As a mathematician and logician, your task is to explain the correct answer to a problem. Use a Chain-of-Thought approach to break down the problem and explain the answer step-by-step.

**Instructions**:

1. **Understand and Clarify**: Make sure you
   understand the problem.
2. **Parsing**: Interpret the inputs in the context of
   the problem.
3. **Reasoning**: Reason about the problem and inputs
   rigorously and work out the answer step-by-step.
4. **Deliberation**: Critically consider the arguments
   for the possible answers.
5. **Answer the question**: Decide on the correct
   answer based on careful and extensive analysis.

**Answer Formatting**: Please write your answer as in
   a code block after the word 'ANSWER:' as
ANSWER:
```
[REPLACE THIS WITH THE CORRECT ANSWER]
```

# For example:

## Example Prompt:
Output the sum of a list of numbers.
The first line is the length of the list, the next
   line is the space-delimited list of numbers.

What is the correct output on input:
```
2
1 2
```

## Example Completion:
ANSWER:
```
3
```

**Problem**:

PROBLEM_DESCRIPTION

TEST_INPUTS

Should the output be
```
EXPECTED_OUTPUT
```
or
```
ACTUAL_OUTPUT
```
or something else?

(The correct answer is likely one of the listed
   options. But if you think that another answer is
   correct, output that as the answer instead.)

```
**Instructions**:
1. **Understand and Clarify**: Make sure you
   understand the problem.
2. **Parsing**: Interpret the inputs in the context of
   the problem.
3. **Reasoning**: Reason about the problem and inputs
   rigorously and work out the answer step-by-step.
4. **Deliberation**: Critically consider the arguments
   for the possible answers.
5. **Answer the question**: Decide on the correct
   answer based on careful and extensive analysis.

Analyze step-by-step, using a Chain-of-Thought
   approach.
Remember to write your answer in a code block after
   the word 'ANSWER:' as
ANSWER:
```

[REPLACE THIS WITH THE CORRECT ANSWER]
```
```

