# OpenReview forum: "ICE-Coder: Integrating White-box and Black-box Testing in Execution-guided Multi-agent Code Generation"
_ICLR.cc/2026/Conference — Submitted to ICLR 2026_

### Official Review · Reviewer_96yT · 2025-10-31

**Soundness:** 2
**Presentation:** 2
**Contribution:** 2
**Rating:** 2
**Confidence:** 4

**Summary:**

This paper provides a multi-agent system to enhance capability of solving complex code generation tasks. It combines the benefits of both single engineer thinking and development team strategy to further help the system. Also, the authors try to integrate the coverage-based testing and code review process to strengthen testing stage. Lastly, they use a prompting-based method to validate the test case correctness.

**Strengths:**

1. this proposed framework is a more engineering-realistic testing loop than previous work.
2. the workflow is clear and easy to understand.
3. it can beat the SOTA with very impressive improvement.
4. its coverage metric is good to measure the quality of tests.
5. the results seem like nearly all combination of model providers and sizes are better than the original ones.

**Weaknesses:**

1. like what the authors claimed, the paper only has results on single LiveCodeBench-Hard dataset. it largely limitate the confidence and reproducibility. Hope the authors can provide more results during the rebuttal process, which can make the submission better.

2. No results or analysis for demonstrating the proposed prompt-based method is better than other methods in validating test case correctness.

3. problem re-writing for baselines induces bias and may change the fairness also.

4. need more analysis of how to avoid the negative improvement of small model like GPT 4.1 NANO. Authors can try to test more small models and investigate whether their results are similar.

5. as the most significant improvement comes from the initial test generation, it is important to analyze the token usage more deeply. maybe adding some examples can help the reader to understand.

**Questions:**

please refer to the weaknesses.

---

> ### Author Response · Authors · 2025-12-03
>
> Thank you for the comments.
>
> 1) Limited testing
>
> Unfortunately, due to limited computing resources, we have not tested it on other benchmarks.
>
> 2) Comparing prompt-based methods to other methods in validating test case correctness
>
> The other methods include 1) not validating test case correctness at all, or 2) statistically determining that, if many codes fail the same test case, it is incorrect. Method 1 is not really feasible --- LLMs tend to produce incorrect test cases, especially for hard problems. Method 2 works on simpler problems, not on harder ones. The problems on LiveCodeBench-Hard are a bit complex to explain in a comment, so we give a toy example. Consider a problem where we have to determine if points `(x1, y1)`, `(x2, y2)` and `(x3, y3)` lie on the same line. A natural solution would be to compare if the slopes `(y1 - y2)/(x1 - x2) == (y1 - y3)/(x1 - x3)` are equal. Suppose for illustration that the LLM produces many codes which use this same, natural method (indeed, the larger, stronger LLMs do not, but this is a simplified example for illustration). An edge case to this would be when `(x1 - x2) == (x1 - x3) == 0` (i.e., the points lie on a vertical line), in which case, these codes would crash. Using a statistical method to evaluate test case correctness would incorrectly discard such valuable test cases
>
> 3) Problem re-writing
>
> AgentCoder, MapCoder etc. only accepted functional problems, which take in the method signature and docstring. Some of the LiveCodeBench problems came with problem descriptions only, where the output should be printed. So it was not possible to have the methods run on an exactly identical problem set. We converted them into a suitable format as follows:
>
> ```
> def solve(inp: str) -> str:
>     """
>     {PROBLEM_DESCRIPTION}
>     """
> ```
>
> where the problem description and test case examples were put in the docstring. We believed this as close as we could get to making them run on an identical dataset. Nevertheless, we put in a disclaimer in the paper that we had to modify the inputs slightly to get the other methods to run.
>
> 4) How to avoid negative improvement for small models
>
> We investigated the negative improvement with GPT 4.1-nano in Appendix D, where we found that the negative improvement was caused by the large initial test set. We propose that small models have difficulty identifying incorrect test cases. Therefore, more targeted tests, such as relying on coverage only or counter-example generation only, lead to better results.
>
> 5) Token usage analysis for initial test set generation
>
> We additionally tried using a small initial test set, which uses a similar number of tokens as our coverage-guided tests. We found that our coverage-guided tests led to better results using fewer tokens.

---

### Official Review · Reviewer_rdR2 · 2025-11-01

**Soundness:** 2
**Presentation:** 2
**Contribution:** 2
**Rating:** 2
**Confidence:** 3

**Summary:**

The paper introduces ICE-CODER, a multi-agent framework for automatic code generation, specifically targeting complex (e.g., competitive programming) problems. The core idea is to improve code reliability by more closely simulating a human software engineering process. The system integrates existing black-box test generation with two key additions: (1) white-box, coverage-guided test generation to find edge cases, and (2) an LLM-based "error diagnosis" step to deliberate on and resolve conflicts between code outputs and test case outputs. The authors evaluate ICE-CODER on the LiveCodeBench-Hard dataset, demonstrating significant improvements over a baseline and claiming state-of-the-art performance by solving 72 out of 90 problems.

**Strengths:**

1. The paper identifies a weakness in current code generation agents: their inability to handle subtle edge cases in complex problems. The motivation to move beyond simple black-box testing is strong and well-founded.
2. The core idea of integrating white-box, coverage-guided testing is a logical and intuitive next step for improving test suite quality. Simulating a "deliberation" or "diagnosis" step to handle faulty tests is also a practical solution to a known problem (LLMs generating incorrect test expectations).

**Weaknesses:**

1. The primary weakness of this paper is its lack of a core, novel contribution. The work appears to be a complex engineering pipeline that skillfully integrates several existing techniques. The paper itself cites prior work on multi-agent code generation (e.g., AgentCoder), LLMs for coverage-guided testing (e.g., Pan et al., 2025; Pizzorno & Berger, 2025), and using execution traces for debugging (e.g., Zhong et al., 2024). The main contribution is this specific combination, which feels more like an incremental "system-building" paper than a fundamental research advance.
2. The experimental support is not strong enough for the claims being made.
The entire evaluation rests on a single, 90-problem dataset (LiveCodeBench-Hard). While this is a relevant dataset, it's a very narrow benchmark. It is unknown if this complex, multi-stage pipeline generalizes to other coding benchmarks (e.g., HumanEval, MBPP) or if it might even regress on simpler problems due to its overhead and complexity.
3. The writing is somewhat convoluted. The paper claims to combine "simulating a software engineering environment" (multi-agent) with "simulating an individual developer's mind" (single-agent), which makes the core metaphor of the framework confusing. It is difficult for the reader to disentangle which parts are re-implementations of prior art and what the single, clean takeaway contribution is.

**Questions:**

1. Could you please provide precise details on the exact modifications made to the problem descriptions to allow AgentCoder, MapCoder, etc., to run? This seems like a critical confounder. How can we be sure that these modifications did not make the task easier for ICE-Coder or harder for the other methods? A fair comparison would require all methods to run on the identical, unmodified problem set.
2. Why was the evaluation limited only to LiveCodeBench-Hard? The authors' hypothesis is that this method is good for complex problems, but this does not excuse the omission of standard benchmarks.
3. Could the authors more clearly articulate what the primary conceptual contribution is, beyond the successful integration and engineering of these known techniques?

---

> ### Author Response · Authors · 2025-12-03
>
> Thank you for the comments.
>
> 1) Modifications made to the problem descriptions to allow other agents to run
>
> AgentCoder, MapCoder etc. only accepted functional problems, which take in the method signature and docstring. Some of the LiveCodeBench problems came with problem descriptions only, where the output should be printed. So it was not possible to have the methods run on an exactly identical problem set. We converted them into a suitable format as follows:
> ```
> def solve(inp: str) -> str:
>     """
>     {PROBLEM_DESCRIPTION}
>     """
> ```
> where the problem description and test case examples were put in the docstring. We believed this as close as we could get to making them run on an identical dataset. Nevertheless, we put in a disclaimer in the paper that we had to modify the inputs slightly to get the other methods to run.
>
> 2) Performance on other benchmarks
>
> Unfortunately, due to limited computing resources, we have not tested it on other, larger benchmarks.
>
> 3) Key contributions
>
> The primary contribution is the use of white-box testing and coverage information in conjunction with code generation. Previous methods relied on black-box testing, or generated code in the same prompt as the test cases. Additionally, we resolve failed tests (whether the code is wrong or the test is wrong) by querying the LLM. Most previous work rely on statistical methods, which we think would discard rare edge cases.

---

### Official Review · Reviewer_mD7x · 2025-11-01

**Soundness:** 2
**Presentation:** 3
**Contribution:** 2
**Rating:** 2
**Confidence:** 2

**Summary:**

The paper presents ICE-coder to improve the code synthesis capabilities of LLMs/LLM-agents. In particular, ICE-coder leverages whitebox test generation techniques to generate tests that can maximum coverage on the source problems. Furthermore, ICE-coder also assess the quality of the tests generated to further improve performance. The authors evaluate ICE-coder on a set of code context problems and found that the approach can improve performance.

**Strengths:**

- deals with an important problem of improving code synthesis capabilities of LLMs/LLM-agents
- the idea of utilizing better whitebox feedback for better test generation approach is interesting

**Weaknesses:**

While I think the overall idea of adding more "white-box" testing methods into agentic-based coding is interesting, I think there are several major weakness to the paper:

- Understanding of what makes coverage useful and important:
	- The paper pushes for a misguided understanding of what makes coverage useful in practice
	- I think we should first acknowledge that "executing code and actually fully functionally testing it are not the same thing"
	- The author writes "When executing the code, we aim for complete statement coverage. This ensures that each statement behaves as expected"
		- I would argue this is a gross simplification of what testing is.
		- Just because you have covered a line of code, it does not mean that line of code is correct in the context of that particular problem
		- The reason why companies utilize basic coverage information is to check that certain features (i.e., new implementations) are at least covered.
		- you will find that often times each line is covered multiple times with good tests written by the developer.
	- Furthermore, in software engineering literature and practice, there are many different ways we can measure coverage and coverage metrics
		- in the paper the authors rely on the most basic metric of line coverage and neglect arguable more useful and aligned coverage metric like path coverage (i.e., the number of execution paths in code which has been coverage)
		- this path coverage would be more aligned with the author's goal of getting more edge testcases (as they may manifest as rare paths)
	- In summary: I think the authors very surface-level of what makes coverage useful in practice in order to demonstrate their technique
	- See [1] for a great article on this topic

- Limited benchmark evaluation and baseline improvement:
	- As pointed out by the authors themselves, the paper only evaluates on less than 100 problems in the LiveCodeBench-hard subset of the problems
	- It is unclear to me why the authors did not evaluate on other code-context benchmarks (or even the medium or easy subset of LiveCodeBench) just to at least show that performance is not degrading
	- Furthermore, from Table 2, ICE-Coder seems to only improve by less than 2 percentage points on 100 problems

minor issues:
- The high number of footnotes are a bit difficult to read, some of the footnotes also appear on the wrong page


[1] Programs, tests, and oracles: the foundations of testing revisited. In Richard N. Taylor, Harald C. Gall, and Nenad Medvidovic, editors, International Conference on Software Engineering, pages 391–400. ACM, 2011.

**Questions:**

1. Can the authors touch on how they use the idea of coverage in their approach? For example why do they only consider the most basic of coverage information (see weakness for more detail)
2. What are the performance of ICE-coder on additional code-context benchmarks including different difficulties of livecodebench problems?

---

> ### Author Response · Authors · 2025-12-03
>
> Thank you for the comments!
>
> 1) Why statement coverage was chosen
>
> Indeed, considering path coverage would yield likely help to explore more edge cases. We chose statement coverage to balance accuracy and efficiency. Admittedly, it would be interesting to try out other coverage metrics.
>
> 2) Performance on other benchmarks
>
> Unfortunately, due to limited computing resources, we have not tested it on other benchmarks.

---

### Official Review · Reviewer_2bhi · 2025-11-01

**Soundness:** 3
**Presentation:** 3
**Contribution:** 2
**Rating:** 6
**Confidence:** 4

**Summary:**

This paper introduces ICE-Coder, an LLM-based multi-agent coding framework designed to improve the reliability of automatically generated code on complex programming tasks. Building on existing multi-agent code generation methods that emulate collaborative software engineering, ICE-Coder integrates insights from code coverage analysis and code review practices to generate white-box tests that complement traditional black-box testing. The approach is evaluated on the LiveCodeBench-Hard benchmark, which solves 72 out of 90 problems, outperforming the baseline of 55.

**Strengths:**

The paper is well written and easy to understand.

The proposed approach is demonstrated to have very good performance in code generation.

**Weaknesses:**

Unclear novelty:

The paper proposes using code coverage features to simulate white-box testing; however, the use of coverage information to facilitate test generation has been extensively explored in prior work, as also acknowledged in the related work section. While the proposed pipeline consists of several steps, it remains unclear what the overall conceptual or methodological novelty is beyond existing approaches, even though the final results are promising.

Limited evaluation:

The experimental evaluation relies solely on a single benchmark, LiveCodeBench-Hard. It is uncertain whether the proposed method generalises to other benchmarks or performs effectively on simpler or more diverse coding tasks. A broader evaluation would strengthen the empirical validity and demonstrate the robustness of the approach.

**Questions:**

Can you further clarify the novelty of this paper?

How does ICE-Coder perform on other benchmarks? How does the difficulty of coding problems affect the effectiveness of the approach?

---

> ### Author Response · Authors · 2025-12-03
>
> Thanks for the comments!
>
> 1) Clarifying the novelty of the paper
>
> While the use of coverage information to generate white-box tests has been explored in prior work, these have largely focused on white-box test generation for existing code-bases, or human-written code. This work generates white-box tests to enhance LLM-generated code.
>
> 2) Performance on other benchmarks
>
> Unfortunately, due to limited computing resources, we have not tested it on other benchmarks.

---

### Meta-Review · Area_Chair_qZmH · 2026-01-05

**Summary:**

This paper received scores of 6, 2, 2, 2, from the four reviewers. The paper introduces a multi-agent coding method that improves black-box testing with coverage-guided test generation to tell whether failure comes from incorrect code or incorrect tests. Results on LiveCodeBench looks promising. However, reviewers think that the novelty is not unclear. Author's response that existing work using coverage information only targets at human-written codes does not offer a convincing rebuttal. Further, experiments are not sufficient, pointed out by reviewers. Authors didn't provide more results due to limited time and resources. Based on these considerations, AC recommends reject.

**Reviewer Concerns:**

Novelty and experiments. Details see meta-review.

**Reviewer Scores:**

I don't think reviewers would've improved the scores even if authors had had more time for rebuttal.

---

### Decision · Program_Chairs · 2026-01-26

Reject